# RestoreGrad: Signal Restoration Using Conditional Denoising Diffusion Models with Jointly Learned Prior

## Abstract

Denoising diffusion probabilistic models (DDPMs) estimate the data distribution by sequentially denoising samples drawn from a prior distribution, which is typically assumed to be the standard Gaussian for simplicity. Owing to their capabilities of generating high-fidelity samples, DDPMs can be utilized for signal restoration tasks in recovering a clean signal from its degraded observation(s), by conditioning the model on the degraded signal. The degraded signals are themselves contaminated versions of the clean signals; due to this correlation, they may encompass certain useful information about the target clean data distribution. However, naively adopting the standard Gaussian as the prior distribution in turn discards such information. In this paper, we propose to improve conditional DDPMs for signal restoration applications by leveraging a more informative prior that is jointly learned with the diffusion model. The proposed framework, called RestoreGrad, exploits the correlation between the degraded and clean signals to construct a better prior for restoration tasks. In contrast to existing DDPMs that just settle on using pre-defined or handcrafted priors, RestoreGrad learns the prior jointly with the diffusion model. To this end, we first derive a new objective function from a modified evidence lower bound (ELBO) of the data log-likelihood, to incorporate the prior learning process into conditional DDPMs. Then, we suggest a corresponding joint learning paradigm for optimizing the new ELBO. Notably, RestoreGrad requires minimum modifications to the diffusion model itself; thus, it can be flexibly implemented on top of various conditional DDPM-based signal restoration models. On speech and image restoration tasks, we show that RestoreGrad demonstrates faster convergence (5-10 times fewer training steps) to achieve on par or better perceptual quality of restored signals over existing DDPM baselines, along with improved robustness to using fewer sampling steps in inference time (2-2.5 times fewer steps), advocating the advantages of leveraging jointly learned prior for efficiency improvements in the diffusion process.

## 1 Introduction

Denoising diffusion probabilistic models (DDPMs) (Ho et al., 2020; Sohl-Dickstein et al., 2015) are latent variable generative models that have shown impressive results in various generative modeling tasks. DDPMs typically consist of i) the *forward process*, where the original data samples are gradually corrupted by adding Gaussian noise to eventually become a standard normal prior; ii) the *reverse process*, in which a neural network model is responsible for recovering the original data samples from the corrupted data by learning to sequentially reverse the diffusion process. Thanks to their exceptional capabilities of generating high-quality data, DDPMs can be applied to various signal restoration tasks – recovering the missing components in a signal due to contamination (e.g., audio recorded with environmental noise (Lu et al., 2021; 2022; Tai et al., 2023b), images obstructed by bad weather conditions (Özdenizci & Legenstein, 2023) or various measurement noises (Croitoru et al., 2023), etc.), by conditioning the DDPM model on the degraded observations.

However, for the diffusion model to adequately learn the reverse process, a large number of training iterations is typically required, leading to potentially slow model convergence. Such inefficiency was recently related to the discrepancy between the real data distribution and the accustomed choice

of the standard Gaussian prior by Lee et al. (2021). They have thus proposed a simple yet effective approach called PriorGrad, which utilizes a data-dependent prior extracted from the conditioner data to construct a better prior noise. Despite demonstrating improved performance on some generative speech tasks, handcrafting a "better" prior from the conditioner would require certain knowledge about the data characteristics, and such guidance may not always exist.

In this paper, our main focus is to investigate the question: **Can we systematically learn a better prior distribution to improve the efficiency of the diffusion generative process, instead of settling on a pre-defined or handcrafted prior?** More specifically, we propose a framework as depicted in Figure 1 at a high level, where the conditional DDPM (parameterized by $\theta$) samples the latent noise $\epsilon$ from a learned prior distribution estimated by another neural network $\psi$, which takes the conditioner

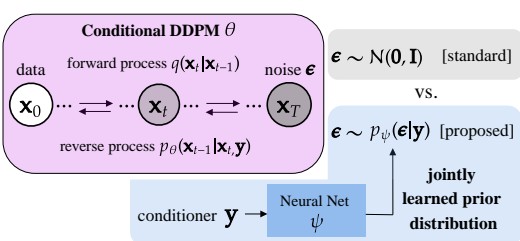

Figure 1: Overview of the proposed method.

$\mathbf{y}$ as input and is jointly trained with $\theta$ to synthesize the data $\mathbf{x}_0$. Currently, traditional DDPMs only incorporate such conditioning information $\mathbf{y}$ in the modeling of the reverse process $p_\theta(\mathbf{x}_{t-1}|\mathbf{x}_t, \mathbf{y})$. The main idea here is, if there is certain correlation between the conditioner $\mathbf{y}$ and the target data $\mathbf{x}_0$, e.g., in signal restoration problems where $\mathbf{y}$ is typically a degraded version of $\mathbf{x}_0$, our framework can exploit such correlation to construct a more informative prior in a systematic manner.

To explore the idea, we introduce RestoreGrad, a new paradigm for improving conditional DDPM by jointly learning the prior distribution, focusing on signal restoration applications. We apply RestoreGrad to speech enhancement (SE) and image restoration (IR) tasks to demonstrate its generality for signals of different nature. For SE, we compare with PriorGrad (Lee et al., 2021) which provides guidance on handcrafting suitable priors in the speech domain. For IR, we show that RestoreGrad serves as a promising solution for improving the baseline DDPM even in a domain that lacks such recipe for handcrafting the prior. As shown in Figure 2, models trained using RestoreGrad are more data and compute-efficient than the baseline DDPM and PriorGrad; they converge faster to achieve higher quality of the restored signal. Further shown in Figure 3, the learned prior is more informative as it better correlates with the desired signal than an isotropic covariance, potentially simplifying the diffusion trajectory for improved efficiency. Our main contributions are summarized as follows:

- We study the problem of learning the prior distribution **jointly** with the conditional DDPM for signal restoration, aiming at providing a more systematic, learning-based treatment to address the inefficiency incurred by existing selections of the prior distribution in DDPM-based methods. In contrast, previous non-standard Gaussian works on DDPMs did not exploit trainable priors, e.g., Nachmani et al. (2021), or were not able to demonstrate the benefits of using learning-based over handcrafted priors for DDPMs, e.g., Lee et al. (2021).

- We propose a new framework called RestoreGrad that learns the prior in conjuncture with the DDPM model through a *prior encoder*, by exploiting the correlation between the targe signal and input degraded signal encoded by an auxiliary *posterior encoder*, for improved model efficiency. Our **two-encoder** learning framework is established based on a novel integration of the evidence lower bounds (ELBOs) of the DDPM and variational autoencoder (VAE) (Kingma & Welling, 2014) to enjoy the advantages of both methodologies.

- Experiments demonstrate that the proposed paradigm is quite general and parameter-efficient, being applicable to DDPM-based signal restoration models for various modalities including images and audio while requiring minimum increase in model complexity.

## 2 BACKGROUND ON DDPMS

**Forward process.** DDPMs (Ho et al., 2020; Sohl-Dickstein et al., 2015) slowly corrupt the training data using Gaussian noise in the forward process. Let $q_{\text{data}}(\mathbf{x}_0)$ be the data density of the original data $\mathbf{x}_0$. The forward process is a fixed Markov Chain that sequentially corrupts the data $\mathbf{x}_0 \sim q_{\text{data}}(\mathbf{x}_0)$ in $T$ diffusion steps, by injecting Gaussian noise according to a variance schedule $\{\beta_t\}_{t=1}^T \in [0, 1)$:

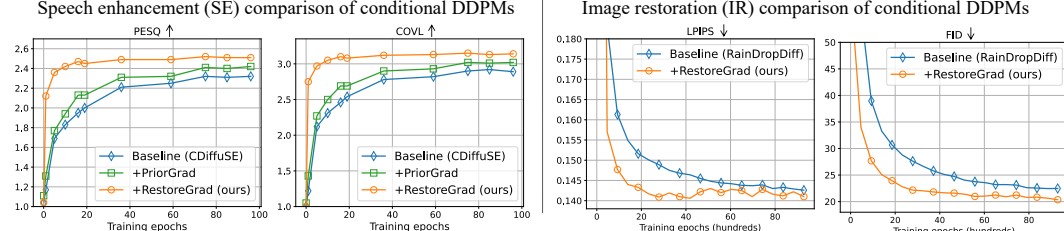

Figure 2: Model learning performance. In speech domain (Left), RestoreGrad outperforms Prior-Grad (Lee et al., 2021), a recently proposed improvement to baseline conditional DDPM (CDiffuSE) by leveraging handcrafted prior. In image domain (Right), RestoreGrad provides a paradigm to improve DDPM baseline (RainDropDiff) where there is no existing recipe for handcrafting the prior.

$$q(\mathbf{x}_{1:T}|\mathbf{x}_0) := \prod_{t=1}^{T} q(\mathbf{x}_t|\mathbf{x}_{t-1}), \quad \text{where} \quad q(\mathbf{x}_t|\mathbf{x}_{t-1}) := \mathcal{N}(\mathbf{x}_t; \sqrt{1-\beta_t}\mathbf{x}_{t-1}, \beta_t\mathbf{I}) \quad (1)$$

is the transition probability at step $t$. It allows the direct sampling of $\mathbf{x}_t$ according to $q(\mathbf{x}_t|\mathbf{x}_0) = \mathcal{N}(\mathbf{x}_t; \sqrt{\bar{\alpha}_t}\mathbf{x}_0, \sqrt{1-\bar{\alpha}_t}\mathbf{I})$, where $\bar{\alpha}_t := \prod_{i=1}^{t} \alpha_i$ with $\alpha_t := 1-\beta_t$. Thus, the sampling can be done as $\mathbf{x}_t = \sqrt{\bar{\alpha}_t}\mathbf{x}_0 + \sqrt{1-\bar{\alpha}_t}\boldsymbol{\epsilon}$, where $\boldsymbol{\epsilon} \sim \mathcal{N}(\mathbf{0}, \mathbf{I})$. A notable assumption is that with a carefully designed variance schedule $\beta_t$ and large enough $T$, such that $\bar{\alpha}_T$ is sufficiently small, $q(\mathbf{x}_T|\mathbf{x}_0)$ converges to $\mathcal{N}(\mathbf{x}_T; \mathbf{0}, \mathbf{I})$ so that the distribution of $\mathbf{x}_T$ is well approximated by the standard Gaussian.

**Reverse process.** One can generate new data samples from $q_{\text{data}}(\mathbf{x}_0)$ by reversing the predefined forward process utilizing the same functional form. More specifically, starting from a noise sample $\mathbf{x}_T \sim p(\mathbf{x}_T) = \mathcal{N}(\mathbf{x}_T; \mathbf{0}, \mathbf{I})$, we can progressively transform the prior noise back into the data by approximating the reverse of the forward transition probability. This process is defined by the joint distribution $p_\theta(\mathbf{x}_{0:T})$ of a Markov Chain with learned Gaussian denoising (i.e., reverse) transitions:

$$p_\theta(\mathbf{x}_{0:T}) := p(\mathbf{x}_T) \prod_{t=1}^{T} p_\theta(\mathbf{x}_{t-1}|\mathbf{x}_t), \quad \text{where} \quad p_\theta(\mathbf{x}_{t-1}|\mathbf{x}_t) := \mathcal{N}(\mathbf{x}_{t-1}; \boldsymbol{\mu}_\theta(\mathbf{x}_t, t), \boldsymbol{\Sigma}_\theta(\mathbf{x}_t, t)) \quad (2)$$

is the reverse of the forward transition probability, parameterized using a deep neural network $\theta$.

**DDPM learning framework.** In an ideal scenario, we would train the model $\theta$ with a maximum likelihood objective such that $p_\theta(\mathbf{x}_0)$ is as large as possible. However, $p_\theta(\mathbf{x}_0)$ is intractable because we have to marginalize over all the possible reverse trajectories to compute it. To circumvent such difficulty, DDPMs (Ho et al., 2020) instead maximize an ELBO of the data log-likelihood, by introducing a sequence of hidden variables $\mathbf{x}_{1:T}$ and the approximate variational distribution $q(\mathbf{x}_{1:T}|\mathbf{x}_0)$:

$$\log p_\theta(\mathbf{x}_0) = \log \int p_\theta(\mathbf{x}_{0:T}) d\mathbf{x}_{1:T} \geq \mathbb{E}_{q(\mathbf{x}_{1:T}|\mathbf{x}_0)} \left[ \log \frac{p_\theta(\mathbf{x}_{0:T})}{q(\mathbf{x}_{1:T}|\mathbf{x}_0)} \right]. \quad (3)$$

With the above parametric modeling of the forward and reverse processes, the ELBO in (3) suggests training the network $\theta$ such that, at each time step $t$, $p_\theta(\mathbf{x}_{t-1}|\mathbf{x}_t)$ is as close as possible to the true forward process posterior conditioned on $\mathbf{x}_0$ (Luo, 2022; Croitoru et al., 2023), i.e.,

$$q(\mathbf{x}_{t-1}|\mathbf{x}_t, \mathbf{x}_0) = \mathcal{N}(\mathbf{x}_{t-1}; \tilde{\boldsymbol{\mu}}_t(\mathbf{x}_t, \mathbf{x}_0), \tilde{\beta}_t\mathbf{I}), \quad (4)$$

where $\tilde{\boldsymbol{\mu}}_t(\mathbf{x}_t, \mathbf{x}_0) := \frac{\sqrt{\bar{\alpha}_{t-1}}\beta_t}{1-\bar{\alpha}_t}\mathbf{x}_0 + \frac{\sqrt{\alpha_t}(1-\bar{\alpha}_{t-1})}{1-\bar{\alpha}_t}\mathbf{x}_t$ and $\tilde{\beta}_t := \frac{1-\bar{\alpha}_{t-1}}{1-\bar{\alpha}_t}\beta_t$.

Based on using a fixed covariance $\boldsymbol{\Sigma}_\theta(\mathbf{x}_t, t) = \sigma_t^2\mathbf{I}$ (e.g., $\sigma_t^2 = \tilde{\beta}_t$) as in Ho et al. (2020), optimizing (3) corresponds to training a network $\boldsymbol{\mu}_\theta(\mathbf{x}_t, t)$ that predicts $\tilde{\boldsymbol{\mu}}_t(\mathbf{x}_t, \mathbf{x}_0)$. Alternatively, Ho et al. (2020) suggested the following reparameterization to rewrite the mean as a function of noise:

$$\boldsymbol{\mu}_\theta(\mathbf{x}_t, t) = \frac{1}{\sqrt{\alpha_t}} \left( \mathbf{x}_t - \frac{\beta_t}{\sqrt{1-\bar{\alpha}_t}} \boldsymbol{\epsilon}_\theta(\mathbf{x}_t, t) \right). \quad (5)$$

They train a neural network $\boldsymbol{\epsilon}_\theta(\mathbf{x}_t, t)$ to predict the real noise $\boldsymbol{\epsilon}$ and use that to compute the mean as (5). Practically the optimization is carried out by minimizing a simplified training objective:

$$\mathcal{L}_{\text{simple}}(\theta) := \mathbb{E}_{\mathbf{x}_0 \sim q_{\text{data}}(\mathbf{x}_0), \boldsymbol{\epsilon} \sim \mathcal{N}(\mathbf{0},\mathbf{I}), t \sim \mathcal{U}(\{1,...,T\})} \left[ \|\boldsymbol{\epsilon} - \boldsymbol{\epsilon}_\theta(\mathbf{x}_t, t)\|^2 \right], \quad (6)$$

which measures, for a random time step $t$, the distance between the actual noise and estimated noise.

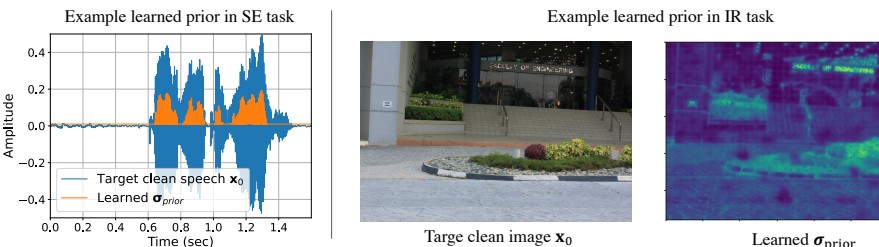

Figure 3: Visualizing the prior distribution learned by RestoreGrad. Here, the assumed form of the prior is a Gaussian: $p_\psi(\epsilon|\mathbf{y}) := \mathcal{N}(\epsilon; \mathbf{0}, \text{diag}\{\sigma_{\text{prior}}^2(\mathbf{y}; \psi)\})$, where $\sigma_{\text{prior}}$ is estimated by the neural network $\psi$ with input $\mathbf{y}$. It appears that $\sigma_{\text{prior}}$ follows the level variation of the speech waveform and preserves the structure of the original image. This indicates that an informative prior approximating the data distribution has been obtained for improved efficiency of the diffusion process.

**Signal restoration by conditional DDPMs.** Signal restoration problems are concerned with recovering the original signals from their degraded observations, which are of paramount importance in reality while remaining challenging, as noises are ubiquitous and may be strong enough to cause significant degradation of the signal quality. Recently, adoption of deep generative models (Kingma & Welling, 2014; Goodfellow et al., 2014; Ho et al., 2020) for signal restoration tasks has considerably increased due to their remarkable capabilities of generating missing components in the data, with conditional DDPMs (Croitoru et al., 2023; Cao et al., 2024) demonstrating substantial promise.

More formally, let $\mathbf{y}$ denote the degraded observation of the clean signal $\mathbf{x}_0$. The task of recovering $\mathbf{x}_0$ given $\mathbf{y}$ by a model $\theta$ can be cast as maximizing the conditional likelihood of data $p_\theta(\mathbf{x}_0|\mathbf{y})$. The problem is in general intractable, but can be approximated by using a DDPM conditioned on $\mathbf{y}$. The main idea is, without modifying the forward diffusion process (1), to learn a conditional diffusion model $\theta$ with $\mathbf{y}$ provided as input to the reverse process (Özdenizci & Legenstein, 2023):

$$p_\theta(\mathbf{x}_{0:T}|\mathbf{y}) := p(\mathbf{x}_T)\prod_{t=1}^{T} p_\theta(\mathbf{x}_{t-1}|\mathbf{x}_t, \mathbf{y}), \quad \text{where} \quad p_\theta(\mathbf{x}_{t-1}|\mathbf{x}_t, \mathbf{y}) := \mathcal{N}(\mathbf{x}_{t-1}; \boldsymbol{\mu}_\theta(\mathbf{x}_t, \mathbf{y}, t), \sigma_t^2\mathbf{I}),$$

such that the sample has high fidelity to the target data distribution conditioned on $\mathbf{y}$. Again, we will consider using the noise estimator network $\epsilon_\theta(\mathbf{x}_t, \mathbf{y}, t)$ instead of predicting the mean $\boldsymbol{\mu}_\theta(\mathbf{x}_t, \mathbf{y}, t)$.

## 3 PROPOSED METHOD

We follow conditional VAEs (Sohn et al., 2015) to maximize the conditional data log-likelihood, $\log p(\mathbf{x}_0|\mathbf{y}) = \log \int p(\mathbf{x}_0, \epsilon|\mathbf{y})d\epsilon$, where $\epsilon$ is an introduced latent variable. To avoid intractable integral, in VAEs an ELBO is utilized as the surrogate objective by introducing an approximate posterior $q(\epsilon|\mathbf{x}_0, \mathbf{y})$. In this work, we propose to further lower bound the ELBO of the VAE by that of the DDPM to incorporate diffusion processes. Specifically, we obtain the lower bound(s) as:

$$\log p(\mathbf{x}_0|\mathbf{y}) \geq \underbrace{\mathbb{E}_{q(\epsilon|\mathbf{x}_0,\mathbf{y})}\left[\log p(\mathbf{x}_0|\mathbf{y}, \epsilon)\right]}_{\text{reconstruction term}} - \underbrace{D_{\text{KL}}\left(q(\epsilon|\mathbf{x}_0, \mathbf{y})||p(\epsilon|\mathbf{y})\right)}_{\text{prior matching term}}$$

$$\geq \mathbb{E}_{q_\phi(\epsilon|\mathbf{x}_0,\mathbf{y})}\left[\underbrace{\mathbb{E}_{q(\mathbf{x}_{1:T}|\mathbf{x}_0)}\left[\log \frac{p_\theta(\mathbf{x}_{0:T}|\mathbf{y}, \epsilon)}{q(\mathbf{x}_{1:T}|\mathbf{x}_0)}\right]}_{\text{conditional DDPM}}\right] - D_{\text{KL}}\big(\underbrace{q_\phi(\epsilon|\mathbf{x}_0, \mathbf{y})}_{\text{Posterior Net}} || \underbrace{p_\psi(\epsilon|\mathbf{y})}_{\text{Prior Net}}\big). \quad (7)$$

The first inequality comes similarly as the ELBO used in (conditional) VAEs (Esser et al., 2018; Luo, 2022) via Jensen's inequality. It consists of the *reconstruction* and *prior matching* terms, which are typically realized by an encoder-decoder architecture with $\epsilon$ being the bottleneck representations.

Our novelty comes with introducing the second inequality to explore the new idea of utilizing the conditional DDPM $\theta$ as the *decoder* module of the VAE. Specifically, **we propose to subsequently lower bound $\log p(\mathbf{x}_0|\mathbf{y}, \epsilon)$ in the reconstruction term of the first inequality, by introducing a sequence of hidden variables $\mathbf{x}_{1:T}$, parameterized by a conditional DDPM $\theta$:**

$$\log p_\theta(\mathbf{x}_0|\mathbf{y}, \epsilon) = \log \int p_\theta(\mathbf{x}_{0:T}|\mathbf{y}, \epsilon)d\mathbf{x}_{1:T} \geq \mathbb{E}_{q(\mathbf{x}_{1:T}|\mathbf{x}_0)}\left[\log \frac{p_\theta(\mathbf{x}_{0:T}|\mathbf{y}, \epsilon)}{q(\mathbf{x}_{1:T}|\mathbf{x}_0)}\right], \quad (8)$$

Figure 4: Proposed RestoreGrad. During training, the conditional DDPM $\theta$, Prior Net $\psi$, and Posterior Net $\phi$ are jointly optimized by (10). During inference, the DDPM $\theta$ samples the latent noise $\epsilon$ from the jointly learned prior distribution to synthesize the clean signal. (Details see Appendix B.1.)

which leads to the second inequality in (7). Moreover, in realization of the *prior matching term*, we propose to parameterize the prior and posterior distributions with two encoder modules, *Prior Net* $\psi$ and *Posterior Net* $\phi$, respectively. This design is inspired by Kohl et al. (2018) for image segmentation with traditional U-Nets. We introduce the idea to DDPM-based signal restoration, where the Posterior Net is exclusively used in training to help the Prior Net learn a more informative prior, by "pulling" the posterior distribution (which encodes richer information of the target distribution by exploiting the correlation between $\mathbf{x}_0$ and $\mathbf{y}$) and the prior distribution towards each other.

Based on (7), we introduce the modified ELBO for the training objective of RestoreGrad:

**Proposition 1** (RestoreGrad). *Assume the prior and posterior distributions are both zero-mean Gaussian, parameterized as $p_\psi(\epsilon|\mathbf{y}) = \mathcal{N}(\epsilon; \mathbf{0}, \Sigma_{prior}(\mathbf{y}; \psi))$ and $q_\phi(\epsilon|\mathbf{x}_0, \mathbf{y}) = \mathcal{N}(\epsilon; \mathbf{0}, \Sigma_{post}(\mathbf{x}_0, \mathbf{y}; \phi))$, respectively, where the covariances are estimated by the Prior Net $\psi$ (taking $\mathbf{y}$ as input) and Posterior Net $\phi$ (taking both $\mathbf{x}_0$ and $\mathbf{y}$ as input). Let us simply use $\Sigma_{prior}$ and $\Sigma_{post}$ hereafter to refer to $\Sigma_{prior}(\mathbf{y}; \psi)$ and $\Sigma_{post}(\mathbf{x}_0, \mathbf{y}; \phi)$ for concise notation. Then, with the direct sampling property in the forward path $\mathbf{x}_t = \sqrt{\bar{\alpha}_t}\mathbf{x}_0 + \sqrt{1 - \bar{\alpha}_t}\epsilon$ at arbitrary timestep $t$ where $\epsilon \sim q_\phi(\epsilon|\mathbf{x}_0, \mathbf{y})$, and assuming the reverse process has the same covariance as the true forward process posterior conditioned on $\mathbf{x}_0$, by utilizing the conditional DDPM $\epsilon_\theta(\mathbf{x}_t, \mathbf{y}, t)$ as the noise estimator of the true noise $\epsilon$, we have the modified ELBO associated with (7):*

$$-ELBO = \underbrace{\frac{\bar{\alpha}_T}{2}\mathbb{E}_{\mathbf{x}_0}\|\mathbf{x}_0\|^2_{\Sigma_{post}^{-1}} + \frac{1}{2}\log|\Sigma_{post}|}_{\textit{Latent Regularization (LR) terms}} + \underbrace{\sum_{t=1}^{T}\gamma_t\mathbb{E}_{(\mathbf{x}_0,\mathbf{y}),\epsilon\sim\mathcal{N}(\mathbf{0},\Sigma_{post})}\|\epsilon - \epsilon_\theta(\mathbf{x}_t, \mathbf{y}, t)\|^2_{\Sigma_{post}^{-1}}}_{\textit{Denoising Matching (DM) terms}}$$

$$+ \underbrace{\frac{1}{2}\Big(\log\frac{|\Sigma_{prior}|}{|\Sigma_{post}|} + tr(\Sigma_{prior}^{-1}\Sigma_{post})\Big)}_{\textit{Prior Matching (PM) terms}} + C, \quad where \ \gamma_t = \begin{cases} \frac{\beta_t^2}{2\sigma_t^2\alpha_t(1-\bar{\alpha}_t)}, & t > 1 \\ \frac{1}{2\alpha_1}, & t = 1 \end{cases}$$

(9)

*are weighting factors, $\|\mathbf{x}\|^2_{\Sigma^{-1}} = \mathbf{x}^T\Sigma^{-1}\mathbf{x}$, $\sigma_t^2 = \frac{1-\bar{\alpha}_{t-1}}{1-\bar{\alpha}_t}\beta_t$ and $C$ is some constant not depending on the learnable parameters $\theta$, $\phi$, and $\psi$.*

The derivation (see Appendix A) is based on combining the conditional VAE and the results in Lee et al. (2021). **Notably, we join the conditional DDPM with the posterior/prior encoders and optimize all modules at once, by connecting the DDPM prior space with the latent space estimated by the encoders.** To this end, the sampling of $\epsilon \sim q_\phi(\epsilon|\mathbf{x}_0, \mathbf{y})$ is performed by the standard reparameterization trick as in VAEs, unlocking end-to-end training via gradient descent on the obtained loss terms explained below:

- *Latent Regularization (LR)* terms: help learn a reasonable prior latent space; e.g., minimizing $\log|\Sigma_{\text{post}}|$ avoids $\Sigma_{\text{post}}$ from becoming arbitrary large due to the presence of its inverse in the weighted norms. These terms can be important for stability reasons in our learnable prior schemes.

- *Denoising Matching (DM)* terms: responsible for training the DDPM to predict the prior noise.

- *Prior Matching (PM)* terms: attempt to find a desirable latent space by agreeing the prior and posterior distributions. Note that we model the distributions as zero-mean Gaussians, exploiting the fact that signals (e.g., waveforms, image pixels) can be properly normalized to zero mean.

**Training of RestoreGrad.** With the the conditional DDPM $\theta$, Prior Net $\psi$, and Posterior Net $\phi$ defined in Proposition 1, we are ready to perform optimization on learning the model parameters of $\theta, \psi, \phi$ based on the modified ELBO. The RestoreGrad framework jointly trains the three neural network modules by minimizing (9). Following existing DDPM literature, we approximate the objective by dropping the weighting constant $\gamma_t$ of the DM terms, leading to the simplified loss:

$$\min_{\theta, \phi, \psi} \; \eta \big( \underbrace{\bar{\alpha}_T \|\mathbf{x}_0\|^2_{\mathbf{\Sigma}_{\text{post}}^{-1}} + \log|\mathbf{\Sigma}_{\text{post}}|}_{\mathcal{L}_{\text{LR}}} \big) + \underbrace{\|\boldsymbol{\epsilon} - \boldsymbol{\epsilon}_\theta(\mathbf{x}_t, \mathbf{y}, t)\|^2_{\mathbf{\Sigma}_{\text{post}}^{-1}}}_{\mathcal{L}_{\text{DM}}} + \lambda \underbrace{\big( \log \frac{|\mathbf{\Sigma}_{\text{prior}}|}{|\mathbf{\Sigma}_{\text{post}}|} + \text{tr}(\mathbf{\Sigma}_{\text{prior}}^{-1} \mathbf{\Sigma}_{\text{post}}) \big)}_{\mathcal{L}_{\text{PM}}},$$

$$(10)$$

where we approximate the expectations by randomly sampling $(\mathbf{x}_0, \mathbf{y}) \sim q_{\text{data}}(\mathbf{x}_0, \mathbf{y})$ and $\boldsymbol{\epsilon} \sim \mathcal{N}(\mathbf{0}, \mathbf{\Sigma}_{\text{post}})$, and the summation by sampling $t \sim \mathcal{U}(\{1, \ldots, T\})$ (exploiting the independency due to Markov assumption (Nichol & Dhariwal, 2021)) in each training iteration. We also introduce $\eta > 0$ for the LR terms and $\lambda > 0$ for PM terms, to exert flexible control of the learned latent space.

**Sampling of RestoreGrad.** In applications that RestoreGrad is mainly concerned with, the ground truth signal $\mathbf{x}_0$ is not available in inference time. The conditional DDPM then samples $\boldsymbol{\epsilon} \sim p_\psi(\boldsymbol{\epsilon}|\mathbf{y}) = \mathcal{N}(\mathbf{0}, \mathbf{\Sigma}_{\text{prior}})$ from the Prior Net instead; the Posterior Net is no longer needed.

**Advantages of RestoreGrad over existing adaptive priors.** In the training stage of RestoreGrad, the latent code $\boldsymbol{\epsilon}$ samples from the posterior $q_\phi(\boldsymbol{\epsilon}|\mathbf{x}_0, \mathbf{y})$ which exploits both the ground truth signal $\mathbf{x}_0$ and conditioner $\mathbf{y}$. It is thus more advantageous than existing works on adaptive priors that solely utilize the conditioner $\mathbf{y}$. To support this statement of the benefits brought by posterior information, we can compare RestoreGrad with the one without the Posterior Net during training – i.e.,

$$\min_{\theta, \psi} \; \eta \big( \bar{\alpha}_T \|\mathbf{x}_0\|^2_{\mathbf{\Sigma}_{\text{prior}}^{-1}} + \log|\mathbf{\Sigma}_{\text{prior}}| \big) + \|\boldsymbol{\epsilon} - \boldsymbol{\epsilon}_\theta(\mathbf{x}_t, \mathbf{y}, t)\|^2_{\mathbf{\Sigma}_{\text{prior}}^{-1}}, \tag{11}$$

which basically removes the Posterior Net $\phi$ and only trains the Prior Net $\psi$ and DDPM $\theta$. In this case, both training and testing become the same scheme. We will present results showing that RestoreGrad performs better with Posterior Net than without it, to support its advantages.

## 4 RELATED WORK

**Diffusion model efficiency improvements.** Das et al. (2023) utilized the shortest path between two Gaussians to reduce the number of diffusion steps. Song et al. (2020) generalized DDPMs via a class of non-Markovian diffusion processes, giving rise to implicit models that use much fewer sampling steps. Nichol & Dhariwal (2021) introduced a few simple modifications to improve the log-likelihood and sampling efficiency. Pandey et al. (2022; 2021) combined the VAE with DDPM to achieve high-fidelity generation, by using DDPM to refine VAE-generated samples. Rombach et al. (2022) performed the diffusion process in the lower dimensional latent space of an autoencoder to achieve high-resolution image synthesis, and Liu et al. (2023b) studied using such latent diffusion models for audio. Popov et al. (2021) explored using a text encoder to extract better representations for continuous-time diffusion based text-to-speech generation. More recently, Nielsen et al. (2024) explored using a time-dependent image encoder to parameterize the mean of the diffusion process. Orthogonal to the above improvements, PriorGrad (Lee et al., 2021) and follow-up work (Koizumi et al., 2022) studied utilizing informative prior extracted from the conditioner data for improving learning efficiency. However, they are still sub-optimal when the conditioner data are degraded versions of the target as in signal restoration applications, and only focused on speech-related tasks.

**Diffusion model based signal restoration.** Built on top of the diffusion models for audio generation and synthesis, e.g., Kong et al. (2021); Chen et al. (2020); Leng et al. (2022), many SE models have been proposed. One of the pioneering work may be CDiffuSE (Lu et al., 2022), which introduced conditional DDPMs to the SE task and demonstrated the potential. Other works (Serrà et al., 2022; Welker et al., 2022; Richter et al., 2023; Yen et al., 2023; Lemercier et al., 2023; Tai et al., 2023a) have also attempted to improve SE by exploiting diffusion models. In the vision domain, diffusion models have also demonstrated impressive performance for IR tasks (Li et al., 2023; Zhu et al., 2023; Huang et al., 2024; Luo et al., 2023; Xia et al., 2023; Fei et al., 2023; Hurault et al., 2022; Liu et al., 2023a; Chung et al., 2023b;a; Zhou et al., 2024; Xiao et al., 2024; Zheng et al., 2024). A notable work utilizing conditional DDPMs for IR is Özdenizci & Legenstein (2023) that achieved impressive performance on several benchmark datasets for restoring vision in adverse weather conditions. Our goal is to add to this interesting body of signal restoration work using diffusion models by exploring the idea of jointly learning the prior distribution and diffusion process for improved model efficiency.

## 5 EXPERIMENTS

### 5.1 APPLICATION TO SPEECH ENHANCEMENT (SE)

#### 5.1.1 EXPERIMENTAL SETUP

**Dataset.** We validate the SE performance on the benchmark SE dataset *VoiceBank+DEMAND* (Valentini-Botinhao et al., 2016). The dataset consists of clean speech clips collected from the VoiceBank corpus (Veaux et al., 2013), mixed with ten types of noise profiles from the DEMAND database (Thiemann et al., 2013). Specifically, the training utterances from VoiceBank are artificially contaminated with the noise samples from DEMAND at 0, 5, 10, and 15 dB signal-to-noise ratio (SNR) levels, amounting to 11,572 utterances. The testing utterances are mixed with different noise samples at 2.5, 7.5, 12.5, and 17.5 dB SNR levels, amounting to 824 utterances.

**Evaluation metrics.** We consider: **PESQ:** Perceptual Evaluation of Speech Quality (ITU-T Rec. P.862.2, 2005). **SI-SNR:** Scale-Invariant SNR (Le Roux et al., 2019). **CSIG, CBAK, COVL:** Mean-opinion-score predictors of signal distortion, background-noise intrusiveness, and overall signal quality, respectively (Hu & Loizou, 2007). In all metrics, a higher score indicates better SE.

**Models.** The following models are compared:

- **Baseline DDPM**: We adopt the CDiffuSE (Base) model as the baseline DDPM from Lu et al. (2022), which is based on DiffWave (Kong et al., 2021) with 4.28M learnable parameters.
- **PriorGrad**: We implement the PriorGrad (Lee et al., 2021) on top of CDiffuSE by changing the prior distribution from the standard Gaussian to $\mathcal{N}(\mathbf{0}, \Sigma_\mathbf{y})$, where $\Sigma_\mathbf{y}$ is the covariance of the data-dependent prior computed based on the conditioner $\mathbf{y}$, using the approach for the application to vocoder in Lee et al. (2021).
- **RestoreGrad**: We incorporate Prior Net and Posterior Net on top of CDiffuSE. Both modules adopt the ResNet-20 architect (He et al., 2016) suitably modified to 1-D convolutions for waveform processing, each has only 93K learnable parameters (only 2% of the CDiffuSE model).

**Configurations.** We adopted the basic configurations same as in Lu et al. (2022). The waveforms were processed at 16kHz sampling rate. The number of forward diffusion steps was $T = 50$. The variance schedule was $\beta_t \in [10^{-4}, 0.035]$, linearly spaced. The batch size was 16. The fast sampling scheme in Kong et al. (2021) was used in the reverse processes with $S = 6$ steps to reduce inference complexity. The inference variance schedule was $\beta_t^{\text{infer}} = [10^{-4}, 10^{-3}, 0.01, 0.05, 0.2, 0.35]$. Adam optimizer (Kingma & Ba, 2014) was utilized with a learning rate of $2 \times 10^{-4}$. We set $\eta = 0.1$ and $\lambda = 0.5$ for (10) (we discuss different choices of $(\eta, \lambda)$ in Appendix C.1). The models were trained on one NVIDIA Tesla V100 GPU (32 GB CUDA memory) and finished 96 epochs in 1 day.

#### 5.1.2 RESULTS

**Improved model convergence.** As shown in Figure 2 (test set performance), RestoreGrad shows better convergence behavior over PriorGrad (handcrafted prior) and CDiffuSE (standard Gaussian prior). For example, **PriorGrad reaches 2.4 in PESQ at 96 epochs, whereas RestoreGrad reaches it in (roughly) 10 epochs, indicating a 10× speed-up.** The results suggest that jointly learning the prior distribution can be beneficial for DDPMs.

**Robustness to reduced number of reverse steps in inference.** RestoreGrad can potentially reduce the inference complexity too. In Figure 5, we show how the trained diffusion models tolerate reduction in the number of inference steps. In each model, we trained the network for 96 epochs and then inferenced with $S = 3$ reverse steps to compare with the originally adopted $S = 6$ steps in Lu et al. (2022). The noise schedule for $S = 3$ was $\beta_t^{\text{infer}} = [0.05, 0.2, 0.35]$, a subset of the $S = 6$ schedule that resulted in best performance. We can see that the baseline DDPM is most sensitive to the step reduction, while PriorGrad shows certain tolerance as leveraging a closer-to-data prior distribution. **Finally, RestoreGrad barely degrades with reduced sampling steps, echoing that a better prior has been obtained as it recovers higher fidelity signal even in fewer reverse steps**.

**Comparison to existing waveform-domain generative SE models.** We present in Table 1 more detailed comparison of RestoreGrad with the baseline CDiffuSE. Here, the scores of CDiffuSE were directly taken from the results reported in Lu et al. (2022) where the model has been fully trained for 445 epochs. For PriorGrad and RestoreGrad we report the mean±std computed based on results of

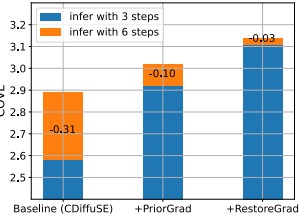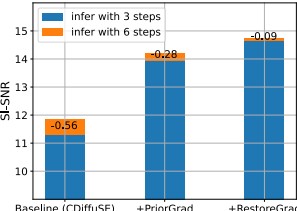

Figure 5: Robustness to the reduction in reverse sampling time steps for inference.

Table 1: Comparison with the fully-trained CDiffuSE model performance reported in Lu et al. (2022). PriorGrad results (trained by ourselves) are also attached for reference.

| Methods | # train epochs | # infer steps | PESQ ↑ | CSIG ↑ | CBAK ↑ | COVL ↑ | SI-SNR ↑ |
|---|---|---|---|---|---|---|---|
| CDiffuSE | 445 | 6 | 2.44 | 3.66 | 2.83 | 3.03 | - |
| + PriorGrad | 96 | 6 | 2.42±3e-3 | 3.67±2e-3 | 2.93±1e-3 | 3.03±2e-3 | 14.21±2e-3 |
| + RestoreGrad (ours) | 96 | 6 | **2.51**±6e-4 | **3.80**±4e-4 | **3.00**±3e-4 | **3.14**±5e-4 | **14.74**±3e-4 |
| | | 3 | 2.50±3e-4 | 3.75±2e-4 | 2.99±2e-4 | 3.11±3e-4 | 14.65±2e-4 |

Table 2: Comparison with existing time-domain, generative SE models.

| Methods | PESQ↑ | CSIG↑ | CBAK↑ | COVL↑ |
|---|---|---|---|---|
| Unprocessed | 1.97 | 3.35 | 2.44 | 2.63 |
| SEGAN | 2.16 | 3.48 | 2.94 | 2.80 |
| DSEGAN | 2.39 | 3.46 | 3.11 | 2.90 |
| SE-Flow | 2.28 | 3.70 | 3.03 | 2.97 |
| DOSE | **2.56** | **3.83** | **3.27** | **3.19** |
| CDiffuSE | 2.44 | 3.66 | 2.83 | 3.03 |
| + RestoreGrad (ours) | 2.51 | 3.80 | 3.00 | 3.14 |

*Best and second best values are indicated with bold text and underlined text, respectively.

10 independent samplings. **We can see that with RestoreGrad applied, the SE model can achieve better performance over the baseline CDiffuSE by only training for 96 epochs (4.6 times lesser than the baseline) in all the metrics**. In addition, halving the number of reverse steps in inference still maintains better performance than the fully-trained CDiffuSE and also the PriorGrad. In Table 2 we also benchmark RestoreGrad with several generative modeling SE approaches: SEGAN (Pascual et al., 2017), DSEGAN (Phan et al., 2020), SE-Flow (Strauss & Edler, 2021), and DOSE (Tai et al., 2023a). Note that although RestoreGrad performs slightly inferior to DOSE, a recent SE model also based on DiffWave (Kong et al., 2021), it was actually achieved with 4.6× fewer epochs than DOSE.

**Does Posterior Net help?** To validate the benefits brought by Posterior Net, we compare the models trained with (11) with the baseline CDiffuSE, PriorGrad, and RestoreGrad models for the SE task in Table 3. For fairness, all models were trained with 96 epochs, inferred with 6 steps. From the results we observe that RestoreGrad achieves better results with Posterior Net than without it, indicating the benefits from being informed of the target $x_0$ by Posterior Net.

Table 3: The better results with Posterior (Post.) Net than without it indicate that exploiting the posterior information during training is helpful.

| SE model | PESQ↑ | COVL↑ | SI-SNR↑ |
|---|---|---|---|
| CDiffuSE (trained for 96 epochs) | 2.32 | 2.89 | 11.84 |
| + PriorGrad | 2.42 | 3.03 | 14.21 |
| + RestoreGrad | **2.51** | **3.14** | **14.74** |
| + RestoreGrad w/o Post. Net ($\eta = 0.01$) | 2.47 | 3.08 | 11.22 |
| + RestoreGrad w/o Post. Net ($\eta = 1$) | 2.48 | 3.12 | 13.29 |

*Best values are indicated with bold text.

## 5.2 APPLICATION TO IMAGE RESTORATION (IR)

### 5.2.1 EXPERIMENTAL SETUP

**Dataset.** Following Özdenizci & Legenstein (2023), we consider the IR task of recovering clean images from their degraded versions contaminated by synthesized noises corresponding to different weather conditions. Two datasets are mainly considered here, where one is a weather-specific dataset called *RainDrop* (Qian et al., 2018) and the other is a multi-weather dataset named *AllWeather* (Valanarasu et al., 2022). The RainDrop dataset consists of images captured with raindrops on the camera sensor which obstruct the view. It has 861 training images with synthetic raindrops, and a test set of 58 images dedicated for quantitative evaluations. The AllWeather dataset is a curated training dataset from Valanarasu et al. (2022), which has 18,069 samples composed of subsets of training images from Snow100K (Liu et al., 2018), Outdoor-Rain (Li et al., 2019) and RainDrop (Qian et al., 2018), in order to create a balanced training set across three weather conditions.

**Evaluation metrics.** Quantitative evaluations between ground truth and restored images are performed via the conventional Peak Signal-to-Noise Ratio (**PSNR**) (Huynh-Thu & Ghanbari, 2008) and Structural SIMilarity (**SSIM**) (Wang et al., 2004), based on the luminance channel Y of the YCbCr color space following Özdenizci & Legenstein (2023), and Learned Perceptual Image Patch Similarity (**LPIPS**)(Zhang et al., 2018) and Fréchet Inception Distance (**FID**) (Heusel et al., 2017).

**Models.** The following IR models are compared:

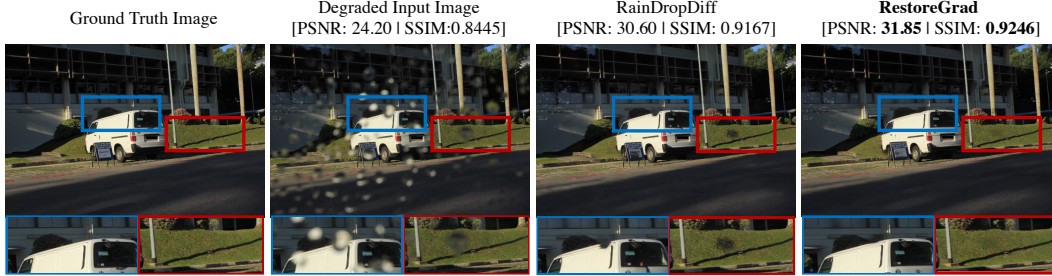

Figure 6: Restored images by RainDropDiff (Özdenizci & Legenstein, 2023) and RestoreGrad (ours) for a test sample from the RainDrop dataset. We provide more examples in Appendix C.2.

Table 4: Weather-specific (RainDrop dataset) model comparison.

| Methods | RainDrop | |
|---|---|---|
| | PSNR ↑ | SSIM ↑ |
| DuRN | 31.24 | 0.9259 |
| RaindropAttn | 31.44 | 0.9263 |
| AttentiveGAN | 31.59 | 0.9170 |
| IDT | 31.87 | 0.9313 |
| RainDropDiff | 32.29 | 0.9422 |
| + RestoreGrad (ours) | **32.69**±0.03 | **0.9441**±7e-5 |

Table 5: Multi-weather model comparison. The models were trained on the AllWeather training set (Valanarasu et al., 2022) and tested on three different weather types.

| Methods | Snow100K-L | | Outdoor-Rain | | RainDrop | |
|---|---|---|---|---|---|---|
| | PSNR ↑ | SSIM ↑ | PSNR ↑ | SSIM ↑ | PSNR ↑ | SSIM ↑ |
| All-in-One | 28.33 | 0.8820 | 24.71 | 0.8980 | 31.12 | 0.9268 |
| TransWeather | 29.31 | 0.8879 | 28.83 | 0.9000 | 30.17 | 0.9157 |
| WeatherDiff | 30.09 | 0.9041 | 29.64 | 0.9312 | 30.71 | 0.9312 |
| + RestoreGrad (ours) | **30.82** | **0.9159** | **30.83** | **0.9411** | **31.78** | **0.9394** |

*Best and second best values are indicated with bold text and underlined text, respectively.

- **Baseline DDPMs**: We consider the RainDropDiff$_{64}$ and WeatherDiff$_{64}$ in Özdenizci & Legenstein (2023) trained on the RainDrop and AllWeather datasets, respectively, as baseline DDPMs. Our work is based on the implementation provided by Özdenizci & Legenstein (2023).
- **RestoreGrad**: We incorporate the additional encoder modules Prior Net and Posterior Net on top of the baseline DDPM. Both encoder modules adopt the ResNet-20 architect (He et al., 2016) with only 0.27M learnable parameters, significantly smaller ($< 0.3\%$) than the baseline DDPM model.

**Configurations.** We used Adam optimizer with a learning rate of $2 \times 10^{-5}$. An exponential moving average with a weight of 0.999 was applied. We used $T = 1000$ and linear noise schedule $\beta_t \in [10^{-4}, 0.02]$, same as Özdenizci & Legenstein (2023). A batch size of 4 was used. The models were trained on two NVIDIA Tesla V100 GPUs of 32 GB CUDA memory and finished training for 9,261 epochs on the RainDrop dataset in 12 days and 887 epochs on the AllWeather dataset in 21 days.

### 5.2.2 RESULTS

**Model convergence:** As presented in Figure 2 (test set performance), RestoreGrad demonstrates faster convergence and better restored image quality over the baseline DDPM (RainDropDiff). For example, **RainDropDiff reaches 32.0 in PSNR at 9.2k epochs, while RestoreGrad reaches it in 1.8k epochs only, indicating a 5× speed-up** due to the effectiveness of the prior learning scheme.

**Image restoration example.** Figure 6 presents examples of the restored images by the models. As can be seen from the images, RestoreGrad is able to better recover the original image, especially in regions of the blue and red boxes where the baseline RainDropDiff fails to remove the rain drop obstructions. The higher PSNR and SSIM scores of RestoreGrad also reflect the improvements.

**Comparison with state-of-the-art IR models.** We compare our method with existing IR models in Table 4, including DuRN (Liu et al., 2019), RaindropAttn (Quan et al., 2019), AttentiveGAN (Qian et al., 2018), and IDT (Xiao et al., 2022). The models were all trained and tested on the RainDrop dataset. The results of the compared models were taken from Özdenizci & Legenstein (2023), where the RainDropDiff was trained for 37,042 epochs. **Our RestoreGrad was only trained for 9,261 epochs (4× fewer than RainDropDiff), and has achieved the highest scores** (here we report mean ± std of RestoreGrad based on results of 10 independent samplings). We also evaluate our method for the multi-weather case in Table 5 with All-in-One (Li et al., 2020) and TransWeather (Valanarasu et al., 2022), where all the models were trained on the AllWeather dataset and tested on the three weather-specific test sets. The numbers of the compared models were taken from Özdenizci & Legenstein (2023), where the WeatherDiff was trained for 1,775 epochs and inferenced with $S = 25$ steps. **Our RestoreGrad was trained for only 887 epochs (2× fewer than WeatherDiff) and inferenced with $S = 10$ steps to already achieve the best performance in all test schemes.**

Table 6: Evaluation on realistic image datasets of IR models trained on synthetic images of AllWeather training set.

| Methods | Gen. | RainDS-Real | Snow-Real |
|---|---|---|---|
| | | NIQE ↓ | NIQE ↓ |
| TransWeather | N | 4.005 | 3.161 |
| WeatherDiff | Y | _3.050_ | **2.985** |
| + RestoreGrad (ours) | Y | **2.556** | _3.015_ |

Table 7: Evaluation of SE models on CHiME-3 test set, where the models were trained on Voice-Bank+DEMAND training set.

| Methods | Gen. | PESQ↑ | CSIG↑ | CBAK↑ | COVL↑ | SI-SNR↑ |
|---|---|---|---|---|---|---|
| Unprocessed | - | 1.27 | 2.61 | 1.93 | 1.88 | 7.51 |
| Demucs | N | 1.38 | 2.50 | 2.08 | 1.88 | - |
| WaveCRN | N | 1.43 | 2.53 | 2.03 | 1.91 | - |
| DOSE | Y | 1.52 | 2.71 | **2.15** | 2.06 | - |
| CDiffuSE | Y | **1.55** | _2.87_ | 2.09 | _2.15_ | _7.67_ |
| + RestoreGrad (ours) | Y | _1.54_ | **2.88** | _2.14_ | **2.16** | **8.45** |

*Best and second best values are indicated with bold text and underlined text, respectively. The column "Gen." indicates if the model is generative (Y) or not (N) in each table.

## 5.3 GENERALIZATION TO OUT-OF-DISTRIBUTION (OOD) AND REALISTIC DATA

We have so far evaluated the models on in-domain scenarios with synthetic noisy data where RestoreGrad has shown substantial improvements. A natural question is that if the demonstrated improvements have actually come at the expense of the model's generalizability to unseen or realistic data. To address the concern, we evaluate the IR models on two additional datasets from Quan et al. (2021); Liu et al. (2018) that consist of real-world images, using the reference-free Natural Image Quality Evaluator (NIQE) metric (Mittal et al., 2012) (a lower score indicates better quality). In Table 6 we see that RestoreGrad is able to perform on par with or better than WeatherDiff and the non-generative TransWeather model. For OOD testing, we evaluate the SE models on the CHiME-3 dataset (Barker et al., 2017) unseen during model training. Table 7 compares RestoreGrad with CDiffuSE that was also trained for 96 epochs, DOSE (Tai et al., 2023a), and two discriminative SE models. We can see that RestoreGrad is able to perform equally well as the CDiffuSE while outperforming DOSE and other non-generative SE models (Demucs (Defossez et al., 2020), WaveCRN (Hsieh et al., 2020)). The results in both tables show that RestoreGrad is capable of improving in-domain performance while maintaining desirable generalization capabilities of generative models.

## 5.4 APPLICATIONS TO OTHER IMAGE RESTORATION TASKS

To demonstrate the generality of our method to benefit conditional DDPMs for IR from other types of degradation, we also perform experiments on image deblurring and super-resolution. We apply RestoreGrad to the baseline conditional DDPM (cDDPM) which implements the same architecture as the patch-based DDPM of Özdenizci & Legenstein (2023) used for weather degradations. Due to space limit, we present results on image deblurring in this section, and leave the discussion on super-resolution to Appendix C.2. For deblurring, we trained the baseline cDDPM and RestoreGrad models and validated their performance on the RealBlur dataset (Rim et al., 2020), a large-scale dataset of real-world blurred images captured both in the camera raw and JPEG formats, leading to two sub-datasets: *RealBlur-R* from the raw images and *RealBlur-J* from the JPEG images. Each training set consists of 3,758 image pairs and each test set consists of 980 image pairs. In Table 8, we present results of the baseline cDDPM and RestoreGrad models trained after 853 epochs. We also include scores of two existing models, SRN-DeblurNet (Tao et al., 2018) and DeblurGAN-v2 (Kupyn et al., 2019), which performed similarly to the baseline cDDPM (taken from results by Rim et al. (2020)), as references for comparison. We can see that, except for LPIPS and FID on RealBlur-J, RestoreGrad is able to achieve improved scores than the baseline cDDPM, and outperform the compared methods.

Table 8: Image deblurring of realistic blurred images.

| Methods | RealBlur-J | | | | RealBlur-R | | | |
|---|---|---|---|---|---|---|---|---|
| | PSNR↑ | SSIM↑ | LPIPS↓ | FID↓ | PSNR↑ | SSIM↑ | LPIPS↓ | FID↓ |
| SRN-DeblurNet | _31.38_ | 0.9091 | - | - | _38.65_ | 0.9652 | - | - |
| DeblurGAN-v2 | 29.69 | 0.8703 | - | - | 36.44 | 0.9347 | - | - |
| Baseline cDDPM | 30.69 | 0.9043 | **0.220** | **15.17** | 37.71 | _0.9777_ | _0.126_ | 14.46 |
| + RestoreGrad (ours) | **31.51** | **0.9095** | _0.224_ | _15.53_ | **38.78** | **0.9796** | **0.122** | **13.61** |

*Bold text for best and underlined text for second best values.

## 6 CONCLUSION

We investigated the potential of jointly learning the prior distribution with the conditional DDPM for improved efficiency. The proposed RestoreGrad provides a more systematic way of estimating the prior than existing selections for diffusion models. Via experiments on SE and IR tasks, we demonstrated the advantages of leveraging learning-based prior with RestoreGrad. A limitation of the current work is that we only focus on signal restoration applications, where we suitably assume a zero-mean Gaussian prior and only learn its covariance. In the future, it can be interesting to research on using a more generic prior form and extend the idea to other modalities and applications.

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

## A    DERIVATION OF PROPOSITION 1

**Proposition 1** (RestoreGrad). *Assume the prior and posterior distributions are both zero-mean Gaussian, parameterized as $p_\psi(\epsilon|\mathbf{y}) = \mathcal{N}(\epsilon; \mathbf{0}, \boldsymbol{\Sigma}_{prior}(\mathbf{y}; \psi))$ and $q_\phi(\epsilon|\mathbf{x}_0, \mathbf{y}) = \mathcal{N}(\epsilon; \mathbf{0}, \boldsymbol{\Sigma}_{post}(\mathbf{x}_0, \mathbf{y}; \phi))$, respectively, where the covariances are estimated by the Prior Net $\psi$ (taking $\mathbf{y}$ as input) and Posterior Net $\phi$ (taking both $\mathbf{x}_0$ and $\mathbf{y}$ as input). Let us simply use $\boldsymbol{\Sigma}_{prior}$ and $\boldsymbol{\Sigma}_{post}$ hereafter to refer to $\boldsymbol{\Sigma}_{prior}(\mathbf{y}; \psi)$ and $\boldsymbol{\Sigma}_{post}(\mathbf{x}_0, \mathbf{y}; \phi)$ for concise notation. Then, with the direct sampling property in the forward path $\mathbf{x}_t = \sqrt{\bar{\alpha}_t}\mathbf{x}_0 + \sqrt{1 - \bar{\alpha}_t}\epsilon$ at arbitrary timestep $t$ where $\epsilon \sim q_\phi(\epsilon|\mathbf{x}_0, \mathbf{y})$, and assuming the reverse process has the same covariance as the true forward process posterior conditioned on $\mathbf{x}_0$, by utilizing the conditional DDPM $\epsilon_\theta(\mathbf{x}_t, \mathbf{y}, t)$ as the noise estimator of the true noise $\epsilon$, we have the modified ELBO associated with (7):*

$$-ELBO = \underbrace{\frac{\bar{\alpha}_T}{2}\mathbb{E}_{\mathbf{x}_0}\|\mathbf{x}_0\|^2_{\boldsymbol{\Sigma}_{post}^{-1}} + \frac{1}{2}\log|\boldsymbol{\Sigma}_{post}|}_{\text{Latent Regularization (LR) terms}} + \underbrace{\sum_{t=1}^{T}\gamma_t\mathbb{E}_{(\mathbf{x}_0,\mathbf{y}),\epsilon\sim\mathcal{N}(\mathbf{0},\boldsymbol{\Sigma}_{post})}\|\epsilon - \epsilon_\theta(\mathbf{x}_t, \mathbf{y}, t)\|^2_{\boldsymbol{\Sigma}_{post}^{-1}}}_{\text{Denoising Matching (DM) terms}}$$

$$+ \underbrace{\frac{1}{2}\big(\log\frac{|\boldsymbol{\Sigma}_{prior}|}{|\boldsymbol{\Sigma}_{post}|} + tr(\boldsymbol{\Sigma}_{prior}^{-1}\boldsymbol{\Sigma}_{post})\big)}_{\text{Prior Matching (PM) terms}} + C, \quad where \ \gamma_t = \begin{cases} \frac{\beta_t^2}{2\sigma_t^2\alpha_t(1-\bar{\alpha}_t)}, & t > 1 \\ \frac{1}{2\alpha_1}, & t = 1 \end{cases}$$

*are weighting factors, $\|\mathbf{x}\|^2_{\boldsymbol{\Sigma}^{-1}} = \mathbf{x}^T\boldsymbol{\Sigma}^{-1}\mathbf{x}$, $\sigma_t^2 = \frac{1-\bar{\alpha}_{t-1}}{1-\bar{\alpha}_t}\beta_t$ and $C$ is some constant not depending on the learnable parameters $\theta$, $\phi$, and $\psi$.*

*Derivation:*

Recall our proposed lower bound in (7) to incorporate the conditional DDPM into the VAE framework is given as:

$$\mathbb{E}_{q_\phi(\epsilon|\mathbf{x}_0,\mathbf{y})}\left[\underbrace{\mathbb{E}_{q(\mathbf{x}_{1:T}|\mathbf{x}_0)}\left[\frac{p_\theta(\mathbf{x}_{0:T}|\mathbf{y},\epsilon)}{q(\mathbf{x}_{1:T}|\mathbf{x}_0)}\right]}_{\mathcal{L}(\theta,\phi)}\right] - D_{\text{KL}}\big(q_\phi(\epsilon|\mathbf{x}_0,\mathbf{y})||p_\psi(\epsilon|\mathbf{y})\big). \tag{12}$$

As assumed in standard DDPMs, the forward diffusion process gradually corrupts the data distribution into the prior distribution, which can be achieved by carefully designing the variance schedule for the forward pass, i.e., $\{\beta_t\}_{t=1}^{T}$, such that $\mathbf{x}_T \to \epsilon$ (as a result of $\bar{\alpha}_T \to 0$). More specifically, the $q(\mathbf{x}_T|\mathbf{x}_0)$ of the forward diffusion process converges in distribution to the approximate posterior $q_\phi(\epsilon|\mathbf{x}_0, \mathbf{y})$ from the posterior encoder $\phi$. Then, the term $\mathcal{L}(\theta, \phi)$ in (12) suggests training a conditional diffusion model $\theta$ to reverse the diffusion trajectory from the estimated distribution of $\epsilon$ given by the posterior encoder $\phi$ back to the target data distribution of $\mathbf{x}_0$.

According to Lee et al. (2021), the form of the loss function $\mathcal{L}(\theta)$ for training the noise estimator network $\theta$ of the conditional DDPM for an arbitrary $\epsilon \sim \mathcal{N}(\mathbf{0}, \boldsymbol{\Sigma})$ is given as:

$$\mathcal{L}(\theta) = \mathcal{L}_0 + \mathcal{L}_T + \sum_{t=2}^{T}\mathcal{L}_{t-1}, \tag{13}$$

where the terms can be explicitly written as:

$$\mathcal{L}_0 := - \mathbb{E}_{q(\mathbf{x}_1|\mathbf{x}_0)} \left[ \log p_\theta(\mathbf{x}_0|\mathbf{x}_1, \mathbf{y}) \right]$$

$$= \frac{1}{2} \log \left( (2\pi\beta_1)^d |\mathbf{\Sigma}| \right) + \frac{1}{2\alpha_1} \mathbb{E}_{\mathbf{x}_0, \boldsymbol{\epsilon} \sim \mathcal{N}(\mathbf{0}, \mathbf{\Sigma})} \| \boldsymbol{\epsilon} - \boldsymbol{\epsilon}_\theta(\mathbf{x}_1, \mathbf{y}, 1) \|^2_{\mathbf{\Sigma}^{-1}},$$

$$\mathcal{L}_{t-1} := \mathbb{E}_{q(\mathbf{x}_t|\mathbf{x}_0)} \left[ \mathcal{D}_{\mathrm{KL}} \left( q(\mathbf{x}_{t-1}|\mathbf{x}_t, \mathbf{x}_0) || p_\theta(\mathbf{x}_{t-1}|\mathbf{x}_t, \mathbf{y}) \right) \right]$$

$$= \frac{\beta_t}{2\alpha_t(1 - \bar{\alpha}_{t-1})} \mathbb{E}_{\mathbf{x}_0, \boldsymbol{\epsilon} \sim \mathcal{N}(\mathbf{0}, \mathbf{\Sigma})} \| \boldsymbol{\epsilon} - \boldsymbol{\epsilon}_\theta(\mathbf{x}_t, \mathbf{y}, t) \|^2_{\mathbf{\Sigma}^{-1}} \tag{14}$$

$$= \frac{\beta_t^2}{2\sigma_t^2 \alpha_t(1 - \bar{\alpha}_t)} \mathbb{E}_{\mathbf{x}_0, \boldsymbol{\epsilon} \sim \mathcal{N}(\mathbf{0}, \mathbf{\Sigma})} \| \boldsymbol{\epsilon} - \boldsymbol{\epsilon}_\theta(\mathbf{x}_t, \mathbf{y}, t) \|^2_{\mathbf{\Sigma}^{-1}},$$

$$\mathcal{L}_T := \mathcal{D}_{\mathrm{KL}} \left( q(\mathbf{x}_T|\mathbf{x}_0) || p(\mathbf{x}_T) \right)$$

$$= \frac{\bar{\alpha}_T}{2} \mathbb{E}_{\mathbf{x}_0} \| \mathbf{x}_0 \|^2_{\mathbf{\Sigma}^{-1}} - \frac{d}{2} (\bar{\alpha}_T + \log(1 - \bar{\alpha}_T)),$$

with $\bar{\alpha}_t := \prod_{i=1}^t \alpha_i$ and $\alpha_t := 1 - \beta_t$ for $t = 1, \ldots, T$ where $\{\beta_t\}_{t=1}^T$ is the noise variance schedule as a hyperparameter, $d$ is the parameter freedom and $\sigma_t^2 = \frac{1 - \bar{\alpha}_{t-1}}{1 - \bar{\alpha}_t} \beta_t$.

In our case, we have assumed modeling of the posterior distribution where the $\boldsymbol{\epsilon}$ is sampled from as the zero-mean Gaussian $\boldsymbol{\epsilon} \sim \mathcal{N}(\mathbf{0}, \mathbf{\Sigma}_{\mathrm{post}})$ where the covariance $\mathbf{\Sigma}_{\mathrm{post}} := \mathbf{\Sigma}_{\mathrm{post}}(\mathbf{x}_0, \mathbf{y}; \phi)$ is estimated by the Posterior Net $\phi$, taking both the ground truth data $\mathbf{x}_0$ and the conditioner $\mathbf{y}$ as input. By directly plugging in $\mathbf{\Sigma} = \mathbf{\Sigma}_{\mathrm{post}}$ for each term in (14), we obtain:

$$\mathcal{L}(\theta, \phi) = \frac{\bar{\alpha}_T}{2} \mathbb{E}_{\mathbf{x}_0} \| \mathbf{x}_0 \|^2_{\mathbf{\Sigma}_{\mathrm{post}}^{-1}} + \frac{1}{2} \log |\mathbf{\Sigma}_{\mathrm{post}}|$$

$$+ \sum_{t=1}^T \gamma_t \mathbb{E}_{(\mathbf{x}_0, \mathbf{y}), \boldsymbol{\epsilon} \sim \mathcal{N}(\mathbf{0}, \mathbf{\Sigma}_{\mathrm{post}})} \| \boldsymbol{\epsilon} - \boldsymbol{\epsilon}_\theta(\underbrace{\sqrt{\bar{\alpha}_t} \mathbf{x}_0 + \sqrt{1 - \bar{\alpha}_t} \boldsymbol{\epsilon}}_{\mathbf{x}_t}, \mathbf{y}, t) \|^2_{\mathbf{\Sigma}_{\mathrm{post}}^{-1}} + C, \tag{15}$$

where

$$\gamma_t = \begin{cases} \frac{\beta_t^2}{2\sigma_t^2 \alpha_t(1 - \bar{\alpha}_t)}, & t > 1 \\ \frac{1}{2\alpha_1}, & t = 1 \end{cases}$$

and $C$ is some constant not depending on the learnable parameters.

For the prior matching term in (12), we can utilize the analytic form of the KL divergence between two Gaussians which leads to:

$$D_{\mathrm{KL}} \left( q_\phi(\boldsymbol{\epsilon}|\mathbf{x}_0, \mathbf{y}) || p_\psi(\boldsymbol{\epsilon}|\mathbf{y}) \right) = \frac{1}{2} \left( \log \frac{|\mathbf{\Sigma}_{\mathrm{prior}}|}{|\mathbf{\Sigma}_{\mathrm{post}}|} + \mathrm{tr}(\mathbf{\Sigma}_{\mathrm{prior}}^{-1} \mathbf{\Sigma}_{\mathrm{post}}) \right), \tag{16}$$

where the covariances $\mathbf{\Sigma}_{\mathrm{prior}} := \mathbf{\Sigma}_{\mathrm{prior}}(\mathbf{y}; \psi)$ and $\mathbf{\Sigma}_{\mathrm{post}} := \mathbf{\Sigma}_{\mathrm{post}}(\mathbf{x}_0, \mathbf{y}; \phi)$.

Combining (15) and (16), we have obtained the $-$ELBO of Proposition 1.

## B  IMPLEMENTATION DETAILS

### B.1  ALGORITHMS

**Algorithm 1:** Training of RestoreGrad

1 **for** $i = 0, 1, 2..., N_{iter}$ **do**
2    Sample $(\mathbf{x}_0, \mathbf{y}) \sim q_{\text{data}}(\mathbf{x}_0, \mathbf{y})$
3    $\boldsymbol{\Sigma}_{\text{prior}} \leftarrow$ Prior Net$(\mathbf{y}; \psi)$
4    $\boldsymbol{\Sigma}_{\text{post}} \leftarrow$ Posterior Net$(\mathbf{x}_0, \mathbf{y}; \phi)$
5    Sample $\boldsymbol{\epsilon} \sim \mathcal{N}(\mathbf{0}, \boldsymbol{\Sigma}_{\text{post}})$ and $t \sim \mathcal{U}(\{1, \dots, T\})$
6    $\mathbf{x}_t = \sqrt{\bar{\alpha}_t}\mathbf{x}_0 + \sqrt{1 - \bar{\alpha}_t}\boldsymbol{\epsilon}$
7    $\mathcal{L}_{\text{LR}} = \bar{\alpha}_T||\mathbf{x}_0||^2_{\boldsymbol{\Sigma}_{\text{post}}^{-1}} + \log|\boldsymbol{\Sigma}_{\text{post}}|$
8    $\mathcal{L}_{\text{DM}} = ||\boldsymbol{\epsilon} - \boldsymbol{\epsilon}_\theta(\mathbf{x}_t, \mathbf{y}, t)||^2_{\boldsymbol{\Sigma}_{\text{post}}^{-1}}$
9    $\mathcal{L}_{\text{PM}} = \log \frac{|\boldsymbol{\Sigma}_{\text{prior}}|}{|\boldsymbol{\Sigma}_{\text{post}}|} + \text{tr}(\boldsymbol{\Sigma}_{\text{prior}}^{-1}\boldsymbol{\Sigma}_{\text{post}})$
10    Update $\theta, \psi, \phi$ with $\nabla_{\theta,\psi,\phi}\ \eta\mathcal{L}_{\text{LR}} + \mathcal{L}_{\text{DM}} + \lambda\mathcal{L}_{\text{PM}}$
11 **end for**

**Algorithm 2:** Sampling of RestoreGrad

1 $\boldsymbol{\Sigma}_{\text{prior}} \leftarrow$ Prior Net$(\mathbf{y}; \psi)$
2 Sample $\mathbf{x}_T \sim \mathcal{N}(\mathbf{0}, \boldsymbol{\Sigma}_{\text{prior}})$
3 **for** $t = T, T - 1, ..., 1$ **do**
4    **if** $t > 0$ **then**
5      Sample $\boldsymbol{\epsilon} \sim \mathcal{N}(\mathbf{0}, \boldsymbol{\Sigma}_{\text{prior}})$
6    **else**
7      $\boldsymbol{\epsilon} = 0$
8    **end if**
9    $\mathbf{x}_{t-1} =$
     $\frac{1}{\sqrt{\alpha_t}}\left(\mathbf{x}_t - \frac{1-\alpha_t}{\sqrt{1-\bar{\alpha}_t}}\boldsymbol{\epsilon}_\theta(\mathbf{x}_t, \mathbf{y}, t)\right) + \sigma_t\boldsymbol{\epsilon}$
10 **end for**
11 **return** $\mathbf{x}_0$

### B.2  EXPERIMENTS ON SPEECH ENHANCEMENT (SE)

#### B.2.1  DATASET

We used the VoiceBank+DEMAND dataset (Valentini-Botinhao et al., 2016) with the same experimental setup as in previous work (Pascual et al., 2017; Phan et al., 2020; Strauss & Edler, 2021; Lu et al., 2022) to perform a direct comparison. The clean speech and noise recordings were provided from the VoiceBank corpus (Veaux et al., 2013) and the Diverse Environments Multichannel Acoustic Noise Database (DEMAND) (Thiemann et al., 2013), respectively, each recorded with sampling rate of 48kHz. Noisy speech inputs used for training were composed by mixing the two datasets with four signal-to-noise ratio (SNR) settings from {0, 5, 10, 15} dB, using 10 types of noise (2 artificially generated + 8 real recorded from the DEMAND dataset) and 28 speakers from the Voice Bank corpus. The test set inputs were made with four SNR settings different from the training set, i.e., {2.5, 7.5, 12.5, 17.5} dB, using the remaining 5 noise types from DEMAND and 2 speakers from the VoiceBank corpus. There are totally 11527 utterances for training and 824 for testing. Note that the speaker and noise classes were uniquely selected for the training and test sets. The dataset is publicly available at: https://datashare.ed.ac.uk/handle/10283/2826. In our experiments, the audio steams were resampled to 16kHz sampling rate.

#### B.2.2  MODEL ARCHITECTURE

**Baseline DDPM-based SE model.** The baseline SE model considered in this work, i.e., CDiffuSE (Lu et al., 2022), performs enhancement in the time domain. We utilized the CDiffuSE base model, which has approximately 4.28M learnable parameters, from the implementation at: https://github.com/neillu23/CDiffuSE. The model is implemented based on DiffWave (Kong et al., 2021), a versatile diffusion probabilistic model for conditional and unconditional waveform generation. The basic model structure of CDiffuSE is similar to that of DiffWave. However, since the target task is SE, CDiffuSE uses the noisy spectral features as the conditioner, rather than the clean Mel-spectral features used in DiffWave utilized for vocoders. After the reverse process is completed, the enhanced waveform further combine the observed noisy signal $\mathbf{y}$ with the ratio 0.2 to recover the high frequency speech in the final enhanced waveform, as suggested in Abd El-Fattah et al. (2008); Defossez et al. (2020).

**PriorGrad.** We implemented the PriorGrad (Lee et al., 2021) on top of the CDiffuSE model by using a data-dependent prior $\mathcal{N}(\mathbf{0}, \boldsymbol{\Sigma}_\mathbf{y})$, where $\boldsymbol{\Sigma}_\mathbf{y}$ is the covariance of the prior distribution computed based on using the mel-spectrogram of the noisy input $\mathbf{y}$. Following the application to vocoder in Lee et al. (2021), we leveraged a normalized frame-level energy of the mel-spectrogram for acquiring data-dependent prior, exploiting the fact that the spectral energy contains an exact correlation to the waveform variance (by Parseval's theorem (Stoica et al., 2005)). More specifically, we computed the frame-level energy by taking the square root of the sum of $\exp(\mathbf{Y})$ over the frequency axis for each time frame, where $\mathbf{Y}$ is the mel-spectrogram of the noisy input $\mathbf{y}$ from the training data. We

then normalized the frame-level energy to a range of $(0, 1]$ to acquire the data-dependent diagonal variance $\Sigma_Y$. Then we upsampled $\Sigma_Y$ in the frame level to $\Sigma_y$ in the waveform-level using the given hop length of computing the mel-spectrogram. We imposed the minimum standard deviation of the prior to 0.1 through clipping to ensure numerical stability during training, as suggested in Lee et al. (2021).

**Prior Net and Posterior Net for RestoreGrad.** The additional encoder modules for the Restore-Grad adopt the ResNet-20 architect (He et al., 2016) using the implementation from: `https://github.com/akamaster/pytorch_resnet_cifar10`. We suitably modified the original 2-D convolutions in ResNet-20 to 1-D convolutions for waveform processing. The modified ResNet-20 model has only 93K learnable parameters (only 2% of the size of CDiffuSE model). The Prior Net takes the noisy speech waveform $\mathbf{y}$ as input, while the Posterior Net takes both the clean and noisy waveforms, $\mathbf{x}_0$ and $\mathbf{y}$, as input, which are concatenated along the channel dimension. We employed the exponential nonlinearity at the network output for estimating the variances of the prior and posterior distributions.

### B.2.3 OPTIMIZATION AND INFERENCE

We used the same configurations of CDiffuSE (Base) (Lu et al., 2022) for optimizing all the models, where the batch size was 16, the Adam optimizer was used with a learning rate of $2 \times 10^{-4}$, and the diffusion steps $T = 50$ with linearly spaced $\beta_t \in [10^{-4}, 0.035]$. For RestoreGrad, we imposed the minimum standard deviation $\sigma_{\min} = 0.1$ by adding it to the output of the Prior Net and Posterior Net to ensure stability during training. The fast sampling scheme in Kong et al. (2021) was used in the reverse processes with $S = 6$ and the inference schedule $\beta_t^{\text{infer}} = [10^{-4}, 10^{-3}, 0.01, 0.05, 0.2, 0.35]$. The models were trained on one NVIDIA Tesla V100 GPU of 32 GB CUDA memory and finished training for 96 epochs in 1 day.

### B.2.4 EVALUATION METRICS

**PESQ:** a speech quality measure using the wide-band version recommended in ITU-T P.862.2 (ITU-T Rec. P.862.2, 2005). It basically models the mean opinion scores (MOS) that cover a scale from 1 (bad) to 5 (excellent). We used the Python-based PESQ implementation from: `https://github.com/ludlows/python-pesq`.

**SI-SNR:** a variant of the conventional SNR measure taking into account the scale-invariance of audio signals. The SI-SDR is a more robust and meaningful metric than the traditional SNR for measuring speech quality. A higher SI-SNR score indicates better perceptual speech quality. We adopted the SI-SNR implementation from: `https://lightning.ai/docs/torchmetrics/stable/audio/scale_invariant_signal_noise_ratio.html`.

**CSIG:** The mean opinion score (MOS) prediction of the signal distortion (from 1 to 5, the higher the better) (Hu & Loizou, 2007). We used the implementation from: `https://github.com/schmiph2/pysepm`.

**CBAK:** MOS prediction of the intrusiveness of background noises (from 1 to 5, the higher the better) (Hu & Loizou, 2007). We used the implementation from: `https://github.com/schmiph2/pysepm`.

**COVL:** MOS prediction of the overall effect (from 1 to 5, the higher the better) (Hu & Loizou, 2007). We used the implementation from: `https://github.com/schmiph2/pysepm`.

### B.3 EXPERIMENTS ON IMAGE RESTORATION (IR)

### B.3.1 DATASETS

We used three standard benchmark image restoration datasets considering adverse weather conditions of snow, heavy rain with haze, and raindrops on the camera sensor, following Özdenizci & Legenstein (2023).

**Snow100K (Liu et al., 2018):** a dataset for evaluation of image desnowing models. We used the test set for evaluation, which consist of 50,000 samples. The images are split into approximately equal sizes of three Snow100K-S/M/L sub-test sets (16,611/16,588/16,801), indicating the synthetic

snow strength imposed via snowflake sizes (light/mid/heavy). The dataset can be downloaded from: `https://sites.google.com/view/yunfuliu/desnownet`.

**Outdoor-Rain (Li et al., 2019):** a dataset of simultaneous rain and fog which exploits a physics-based generative model to simulate not only dense synthetic rain streaks, but also incorporating more realistic scene views, constructing an inverse problem of simultaneous image deraining and dehazing. We used the test set, denoted in Li et al. (2019) as Test1, which is of size 750 for quantitative evaluations. The dataset can be accessed at: `https://github.com/liruoteng/HeavyRainRemoval`.

**RainDrop (Qian et al., 2018):** a dataset of images with raindrops introducing artifacts on the camera sensor and obstructing the view. It consists of 861 training images with synthetic raindrops, and a test set of 58 images dedicated for quantitative evaluations, denoted in Qian et al. (2018) as RainDrop-A. The dataset is provided at: `https://github.com/rui1996/DeRaindrop`.

In addition, we also used the composite dataset for multi-weather IR model training:

**AllWeather (Valanarasu et al., 2022):** is a dataset of 18,069 samples composed of subsets of training images from the training sets of the three datasets above, in order to create a balanced training set across three weather conditions with a similar approach to Li et al. (2020). The dataset is publicly available at: `https://github.com/jeya-maria-jose/TransWeather`.

### B.3.2 MODEL ARCHITECTURE

**Baseline DDPM-based IR models.** The baseline IR models considered in this work, i.e., the Rain-DropDiff and WeatherDiff from Özdenizci & Legenstein (2023), perform patch-based diffusive restoration of the images. The models perform diffusion process at the patch level, where overlapping $p \times p$ patches are taken as input. When sampling, all $p \times p$ patches extracted from the image with a hop size $r$ are processed by the DDPM model, utilizing the mean estimated noise based sampling updates for the overlapping pixels to synthesize the clean image. In this work, we considered $p = 64$ and $r = 16$, which correspond to the RainDropDiff$_{64}$ and WeatherDiff$_{64}$ models (with 110M and 82 M learnable parameters, respectively) provided by the authors at: `https://github.com/IGITUGraz/WeatherDiffusion`.

**Prior Net and Posterior Net for RestoreGrad.** The additional encoder modules for the Restore-Grad adopt the ResNet-20 architect (He et al., 2016) using the implementation from: `https://github.com/akamaster/pytorch_resnet_cifar10`. The ResNet-20 model has 0.27M learnable parameters, which is less than 0.3% of the size of RainDropDiff and WeatherDiff. The Prior Net takes the noisy image $\mathbf{y}$ as input, while the Posterior Net takes both the clean and noisy images, $\mathbf{x}_0$ and $\mathbf{y}$, as input, which are concatenated along the channel dimension. We employed the exponential nonlinearity at the network output for estimating the variances of the prior and posterior distributions.

### B.3.3 OPTIMIZATION AND INFERENCE

We used the same configurations of Özdenizci & Legenstein (2023) for optimizing all the models, except the batch size was 4 instead of 16 due to GPU memory limitation. The Adam optimizer with a fixed learning rate of $2 \times 10^{-5}$ was used for training models without weight decay, and an exponential moving average with a weight of 0.999 was applied during parameter updates. The number of diffusion steps was $T = 1000$ and the noise schedule was $\beta_t \in [10^{-4}, 0.02]$, linearly spaced. For inference, we used $S = 10$ sampling timesteps for each model that we trained on our own. We did not use the deterministic implicit sampling scheme as in Özdenizci & Legenstein (2023) for our RestoreGrad-based DDPM models as we found using the normal stochastic sampling scheme actually works better. The models were trained on 2 NVIDIA Tesla V100 GPU of 32 GB CUDA memory and finished training for 9,261 epochs on the RainDrop dataset in 12 days and 887 epochs on the AllWeather dataset in 21 days.

### B.3.4 EVALUATION METRICS

**PSNR:** a non-linear full-reference metric that compares the pixel values of the original reference image to the values of the degraded image based on the mean squared error (Huynh-Thu & Ghanbari, 2008). A higher PSNR indicates better reconstruction quality of images in terms of distortion.

PSNR can be calculated for the different color spaces. We followed Özdenizci & Legenstein (2023) to compute PSNR based on the luminance channel Y of the YCbCr color space. We used the implementation form `https://github.com/JingyunLiang/SwinIR` for PSNR calculation.

**SSIM:** a non-linear full-reference metric compares the luminance, contrast and structure of the original and degraded image (Wang et al., 2004). It provides a value from 0 to 1, the closer the score is to 1, the more similar the degraded image is to the reference image. We followed Özdenizci & Legenstein (2023) to compute SSIM based on the luminance channel Y of the YCbCr color space. We used the implementation form `https://github.com/JingyunLiang/SwinIR` for SSIM calculation.

**Learned Perceptual Image Patch Similarity (LPIPS)** (Zhang et al., 2018) and **Fréchet Inception Distance (FID)** (Heusel et al., 2017): to provide better quantification of perceptual quality over the traditional distortion measures of PSNR and SSIM (Blau & Michaeli, 2018; Freirich et al., 2021). For the LPIPS we used the implementation from `https://github.com/richzhang/PerceptualSimilarity`, and for FID we used the implementation from `https://github.com/chaofengc/IQA-PyTorch`. In both metrics, a lower score indicates better perceptual quality of the restored image.

### B.4 EXPERIMENTS ON GENERALIZATION TO OOD AND REALISTIC DATA

#### B.4.1 DATASETS

The additional datasets considered for experiments on realistic data for the IR task are:

**RainDS-Real (Qian et al., 2018):** is the raindrop removal test subset of the RainDS dataset presented in Qian et al. (2018). It consists of 98 real-world captured raindrop obstructed images. The dataset is publicly available at: `https://github.com/Songforrr/RainDS_CCN`.

**Snow100K-Real (Liu et al., 2018):** is the subset of the Snow100K dataset (Liu et al., 2018) that consists of 1,329 realistic snowy images for testing real-world restoration cases. The dataset can be accessed at: `https://sites.google.com/view/yunfuliu/desnownet`.

The additional dataset considered for experiments on OOD data of the SE task is:

**CHiME-3 (Barker et al., 2017):** is a 6-channel microphone recording of talkers speaking in a noisy environment, sampled at 16 kHz. It consists of 7138 and 1320 simulated utterances for training and testing, respectively, which are generated by artificially mixing clean speech data with noisy backgrounds of four types, i.e. cafe, bus, street, and pedestrian area. In this paper, we only take the 5-th channel recordings for the experiments. The dataset information can be found at: `https://www.chimechallenge.org/challenges/chime3/data`.

#### B.4.2 EVALUATION METRICS

The additional evaluation metric used in the corresponding section is:

**NIQE:** is a reference-free quality assessment of real-world restoration performance introduced by Mittal et al. (2012) which measures the naturalness of a given image without using any reference image. A lower NIQE score indicates better perceptual image quality. We used the NIQE implementation from: `https://github.com/chaofengc/IQA-PyTorch`.

### B.5 APPLICATIONS TO OTHER IMAGE RESTORATION TASKS

#### B.5.1 DATASETS

The datasets considered for experiments on image deblurring and super-resolution tasks are:

**RealBlur (Rim et al., 2020):** a large-scale dataset of real-world blurred images and ground truth sharp images for learning and benchmarking single image deblurring methods. The images were captured both in the camera raw and JPEG formats, leading to two datasets: *RealBlur-R* from the raw images and *RealBlur-J* from the JPEG images. Each training set consists of 3,758 image pairs and each test set consists of 980 image pairs. The dataset can be downloaded from: `https://cg.postech.ac.kr/research/realblur/`.

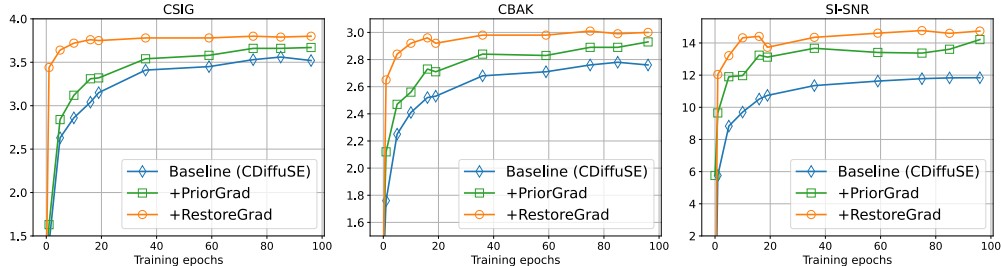

Figure 7: Model learning performance in terms of CSIG, CBAK, and SI-SNR metrics. Improved training behavior of RestoreGrad over CDiffuSE and PriorGrad is observed among all metrics.

**DIV2K (Agustsson & Timofte, 2017; Timofte et al., 2017):** a dataset of 2K resolution high quality images collected from the Internet as part of the NTIRE 2017 super-resolution challenge. There are 800, 100, and 100 images for training, validation, and testing, respectively. The dataset provides $\times 2$, $\times 3$, and $\times 4$ downscaled images with bicubic and unknown downgrading operations. The dataset can be downloaded from: https://data.vision.ee.ethz.ch/cvl/DIV2K/.

### B.5.2 MODEL ARCHITECTURE

The baseline conditional DDPM (cDDPM) implements the same architecture as the patch-based denoising diffusion model of WeatherDiff (Özdenizci & Legenstein, 2023). The Prior Net and Posterior Net of RestoreGrad also adopt the same ResNet models as in the IR experiments under adverse weather conditions. For more details please refer to Appendix B.3.2.

### B.5.3 OPTIMIZATION AND INFERENCE

The models were optimized and inferenced using the same configurations and settings as given in Appendix B.3.3 for the IR experiments under adverse weather conditions. The models were trained on 2 NVIDIA Tesla V100 GPU of 32 GB CUDA memory and finished training for 853 epochs on the RealBlur-{R,J} dataset each in 5 days and 2000 epochs on the DIV2K-{$\times 2, \times 4$} dataset each in 3 days.

## C ADDITIONAL EXPERIMENTAL RESULTS

### C.1 ADDITIONAL RESULTS ON SE

**Model learning performance in terms of other metrics.** In addition to the results evaluated by PESQ and COVL in Figure 2, we provide the learning curves in terms of the CSIG, CBAK, and SI-SNR metrics in Figure 7, to further support the advantages of RestoreGrad over the baseline DDPM and PriorGrad for improved training behavior and efficiency.

**Performance with using different numbers of inference steps.** In Figure 8, we show how the trained diffusion models perform with respect to using different numbers of reverse steps for inference. Specifically, in each case of CDiffuSE, PriorGrad, and RestoreGrad, we trained the model for 96 epochs and then inferenced with $S \in \{3, 4, 5\}$ reverse steps to compare with the originally adopted $S = 6$ steps in Lu et al. (2022). We used $\beta_t^{\text{infer}} = [10^{-4}, 10^{-3}, 0.05, 0.2, 0.35]$ for $S = 5$, $\beta_t^{\text{infer}} = [10^{-4}, 0.05, 0.2, 0.35]$ for $S = 4$, and $\beta_t^{\text{infer}} = [0.05, 0.2, 0.35]$ for $S = 3$. These choices were selected from the subsets of the original noise schedule for $S = 6$, i.e., $\beta_t^{\text{infer}} = [10^{-4}, 10^{-3}, 0.01, 0.05, 0.2, 0.35]$, that resulted in best performance of the models. For the figure we can see that as $S$ becomes smaller, the baseline CDiffuSE degrades considerably, while PriorGrad shows certain resistance, and RestoreGrad manages to maintain the high performance.

We present more comparison in Table 9 in terms of SI-SNR, CSIG, CBAK, and COVL metrics. The results further support that RestoreGrad is much more robust to the reduction in sampling steps, achieving the best quality scores in all the metrics over the baseline DDPM and PriorGrad across all sampling steps considered.

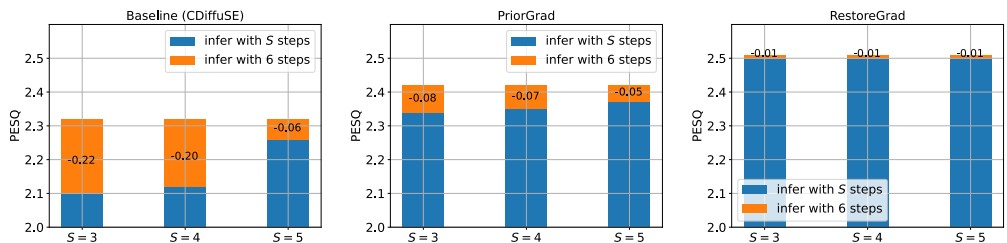

Figure 8: Effect of using reduced numbers of sampling steps in inference on the SE performance, in terms of PESQ. RestoreGrad demstrates strongest endurance to the reduction in reverse sampling steps for inference.

Table 9: Performance comparison of RestoreGrad with the baseline DDPM (CDiffuSE) and Prior-Grad for using various numbers of sampling steps $S$ during inference.

| Methods | SI-SNR ↑ | | | | CSIG ↑ | | | | CBAK ↑ | | | | COVL ↑ | | | |
|---|---|---|---|---|---|---|---|---|---|---|---|---|---|---|---|---|
| | S=6 | S=5 | S=4 | S=3 | S=6 | S=5 | S=4 | S=3 | S=6 | S=5 | S=4 | S=3 | S=6 | S=5 | S=4 | S=3 |
| CDiffuSE (Lu et al., 2022) | 11.84 | 11.46 | 11.32 | 11.28 | 3.52 | 3.44 | 3.15 | 3.13 | 2.76 | 2.72 | 2.64 | 2.63 | 2.89 | 2.82 | 2.60 | 2.58 |
| + PriorGrad (Lee et al., 2021) | 14.21 | 13.98 | 13.93 | 13.93 | 3.67 | 3.61 | 3.56 | 3.54 | 2.93 | 2.90 | 2.88 | 2.88 | 3.02 | 2.97 | 2.93 | 2.92 |
| + RestoreGrad (ours) | **14.74** | **14.66** | **14.64** | **14.65** | **3.80** | **3.77** | **3.75** | **3.75** | **3.00** | **2.99** | **2.99** | **2.99** | **3.14** | **3.12** | **3.11** | **3.11** |

*Best values are indicated with bold text.

**Visualizing the learned prior.** It would be interesting to see how the latent noise prior that has been learned by RestoreGrad looks like and how it compares with that of the PriorGrad. In Figure 9 we present an example of a randomly chosen noisy speech waveform and the corresponding latent noise $\Sigma_y = \text{diag}\{\sigma_y^2\}$ of PriorGrad and that of RestoreGrad (with $(\eta, \lambda) = (0.1, 0.5)$ for (10)). It can be seen that the variances of the pre-defined (PriorGrad) and learned (RestoreGrad) latent noise distributions are actually quite different, though both show the trend of following the variation of the conditioner signal level. This trend indicates that both latent distributions aim to better approximate the true signal distribution in a more informative manner for improved efficiency, as against the standard Gaussian prior used in the original DDPM. Note that in the RestoreGrad training, we have chosen a proper KL weight $\lambda$ so that the Prior Net distribution matches the Posterior Net distribution reasonably well without harming the reconstruction ability of the DDPM model. On the other hand, using a too large $\lambda$ might lead to a collapsed latent space as the optimization could have put too much emphasis on matching the prior and posterior distribution, discarding the contribution of the reconstruction loss term. In contrast, using a too small $\lambda$ might result in large discrepancy between the learned prior and posterior distributions, as also illustrated in Figure 9. Empirically, we found a naive choice of 1 works reasonably well and also for similar values, e.g., 0.5, 10, etc., as similarly observed in the VAE-type model of Kohl et al. (2018).

**Restoration performance vs $\eta$.** An additional hyperparameter introduced in the RestoreGrad objective function (10) is the latent regularization weight $\eta$. An appropriate value of $\eta$ should be large enough to properly regularize the learned latent space for avoiding instability, while not putting too much contribution so that it will not adversely affect the signal reconstruction and prior matching aspects. Empirically, we found the overall SE performance is not very sensitive to the value of $\eta$ across a wide range, as shown in Figure 10: roughly in the range of $[10^{-2}, 10]$ of the $\eta$ value we see that RestoreGrad (here $\lambda$ was fixed at 0.5) gives better (or equally good) results over both PriorGrad and CDiffuSE, indicating that a good $\eta$ is not challenging to find. On the right-hand side of the figure, we also show how the learned latent variances look like if using a too small and a too large $\eta$. We can see that if $\eta$ is too small, it might fail to regularize the latent space properly and result in arbitrary large variances that could lead to degraded performance. On the other hand, if $\eta$ is too large, it might affect the signal reconstruction and prior matching facets, causing the performance to also degrade.

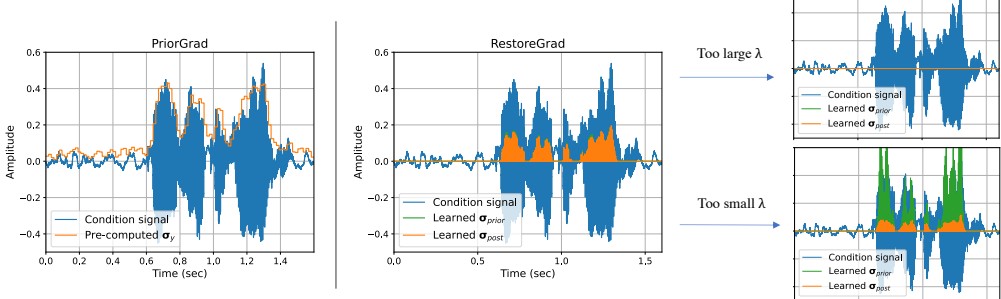

Figure 9: An example of learned latent distribution variances, $\Sigma_{\text{prior}} = \text{diag}\{\boldsymbol{\sigma}^2_{\text{prior}}\}$ and $\Sigma_{\text{post}} = \text{diag}\{\boldsymbol{\sigma}^2_{\text{post}}\}$ by RestoreGrad, and the effect of the KL weight $\lambda$ of the prior matching loss $\mathcal{L}_{\text{PM}}$ on the resulting latent distribution variances. The pre-computed variance of the handcrafted prior using PriorGrad is also presented for reference purposes.

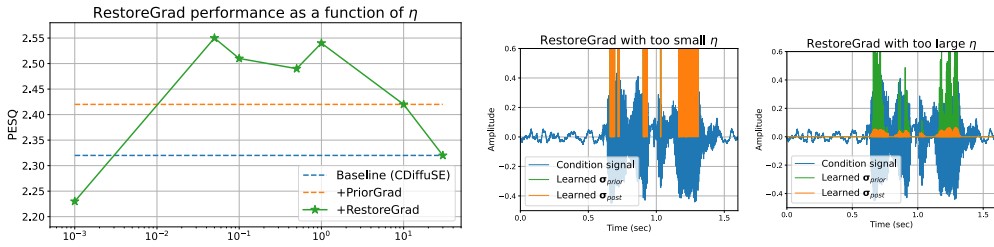

Figure 10: SE performance sensitivity to the latent regularization weight $\eta$ of $\mathcal{L}_{\text{LR}}$.

**Evaluation using automatic speech recognition (ASR).** Following Benita et al. (2024) who perform evaluation of diffusion-based speech generation using ASR, we evaluate the SE model as a front-end denoiser for ASR under noisy environments. To this end, we pre-process the noisy Voice-Band+DEMAND test data samples through the well-trained SE model and feed the denoised audio separately to two pre-trained ASR engines taken from the NVIDIA NeMo toolkit[1]: *Conformer-transducer-large* (Gulati et al., 2020) and *Citrinet-1024* (Majumdar et al., 2021). We report the word error rate (WER) and character error rate (CER) for each ASR engine outcome, where the lower WER / CER indicate better performance. The results are presented in Table 10 with all the SE models trained after 96 epochs, inferred using 6 steps. It is interesting to see that CDiffuSE and PriorGrad actually lead to worse performance than the unprocessed speech case for Citrinet ASR. Our RestoreGrad is able to achieve the lowest WER and CER for both ASR models, demonstrating its efficacy for enhancing machine learning capabilities under noisy environments.

Table 10: Following Benita et al. (2024) who perform evaluation of diffusion-based speech generation using ASR, we evaluate SE models on two ASR engines (Conformer, Citrinet) for the Voice-Band+DEMAND test set. The results further confirm the superiority of RestoreGrad over the baseline and PriorGrad.

| SE model | ASR: WER ↓ (%) / CER ↓ (%) | |
| | Conformer (Gulati et al., 2020) | Citrinet (Majumdar et al., 2021) |
| --- | --- | --- |
| Unprocessed | 6.62 / 6.15 | 8.69 / 6.86 |
| CDiffuSE | 6.55 / 6.01 | 9.77 / 7.41 |
| + PriorGrad | 6.13 / 5.70 | 9.15 / 7.00 |
| + RestoreGrad | **5.07 / 5.27** | **8.15 / 6.51** |

*Best values are indicated with bold text.

[1] https://github.com/NVIDIA/NeMo

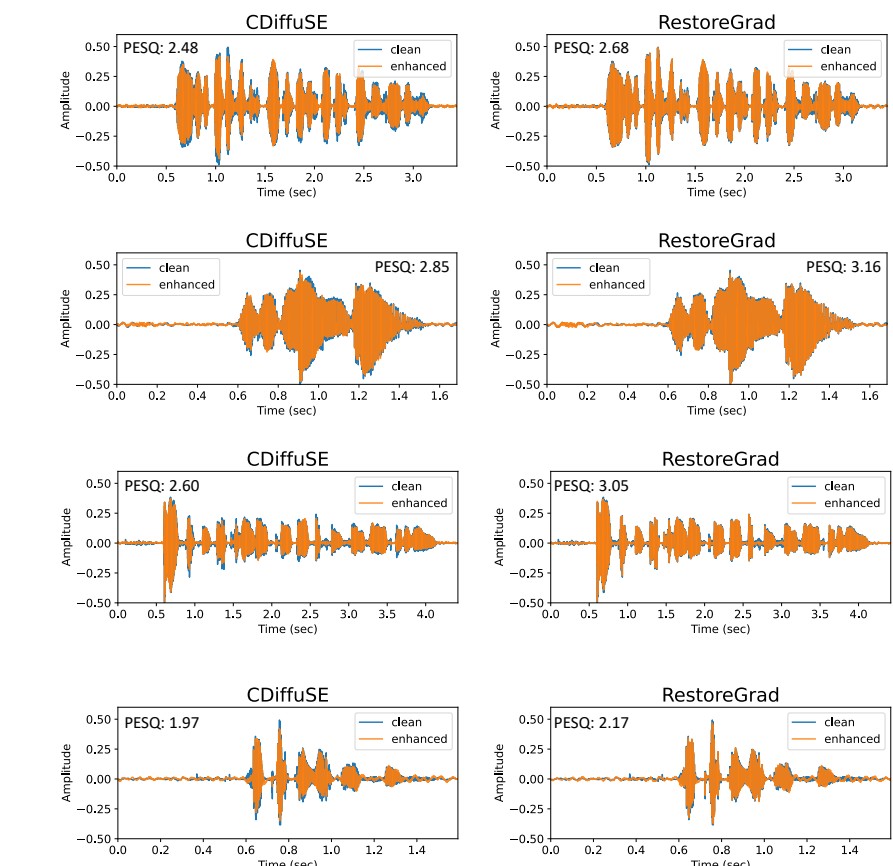

Figure 11: Enhanced speech examples of the baseline DDPM (CDiffuSE) and the proposed Restore-Grad for several noisy samples taken from the VoiceBank+DEMAND test set.

**Enhanced speech examples.** We present several audio examples in Figure 11 to facilitate the comparison of the baseline DDPM and our RestoreGrad. It can be seen the RestoreGrad is able to recover a better speech signal closer to the target clean speech, which is also reflected by the higher PESQ scores obtained. A few more audio samples can be accessed at https://anonymous.4open.science/r/SE_audio_samples-2D7C/.

**SE quality and encoder model size trade-offs.** We have further conducted experiments on using different model sizes for the Prior and Posterior Nets. The results shown in Table 11 clearly show that the restore speech quality improves with increased model size of the encoders (Prior Net and Posterior Net), indicating there is a trade-off between the restoration signal quality and encoder model complexity.

Table 11: SE comparison of RestoreGrad models using three different sizes of the encoder modules (i.e., Prior Net and Posterior Net). *The Base (96K) model is the one used in main experiments.

| Encoder size | PESQ ↑ | COVL ↑ | SSNR ↑ | SI-SNR ↑ |
|---|---|---|---|---|
| Tiny (24K params) | 2.48 | 3.11 | 5.10 | 13.74 |
| Base (96K params) | 2.51 | 3.14 | 5.92 | 14.74 |
| Large (370K params) | 2.54 | 3.16 | 6.15 | 15.01 |

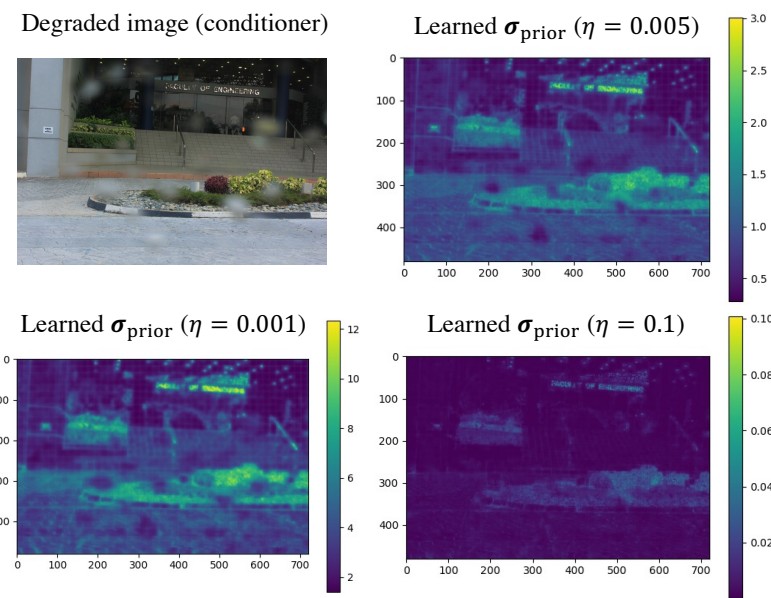

Figure 12: Visualization of learned prior distribution variances with various $\eta$ for a sample image taken from the RainDrop test set (Qian et al., 2018). Mind the magnitude color bar of each figure. We can see that a larger $\eta$ results in smaller variance of the prior distribution, while a smaller $\eta$ leads to larger variance.

## C.2 ADDITIONAL RESULTS ON IR

**Visualizing the learned prior.** We visualize the learned prior distribution variances for a chosen image input with various $\eta$ values in Figure 12 since we are interested in the effect of this newly introduced hyperparameter. We plot the results for the first channel of the image. The original contaminated image (i.e., the conditioner $\mathbf{y}$ to the DDPM model) is also presented for reference purposes. As expected for the latent space regularization effect, a large $\eta$ results in smaller variances as enforcing stronger regularization, while a small $\eta$ leads to larger variances, as observed in the plots. Moreover, the learned prior appears to preserve the structure of the image, indicating that it tends to learn a prior distribution that approximates the data distribution.

**Restoration performance vs. $\eta$ and $\lambda$.** We also study the IR performance of the RestoreGrad models trained across various combinations of $\eta$ and $\lambda$ in Table 12, where the models were trained and tested on the RainDrop dataset. The results show that RestoreGrad works effectively for a wide range of $\eta$ and $\lambda$ values as outperforming the baseline DDPM model, RainDropDiff from Özdenizci & Legenstein (2023), which utilizes the standard Gaussian prior for the diffusion process.

**Experiments on image super-resolution.** We further study the benefits of RestoreGrad over the baseline conditional DDPM (cDDPM) model on image super-resolution tasks with the DIV2K dataset (Agustsson & Timofte, 2017; Timofte et al., 2017). We compare RestoreGrad with the baseline cDDPM model (the same architecture of the patch-based DDPM of WeatherDiff (Özdenizci & Legenstein, 2023)) for ×2 and ×4 downscale factor subsets (with bicubic downgrading operators). There are 800 images for training and 100 images for validation in each subset. For both subsets, we trained a baseline cDDPM and the RestoreGrad models for 2000 epochs on the training set and evaluated their performance on the corresponding validation set. The results are presented in Table 13, where we can see that except for the LPIPS metric, RestoreGrad is more beneficial then the baseline cDDPM in terms of achieving better scores in the other three metrics.

Table 12: RestoreGrad performance for various $\eta$ and $\lambda$, where the models were trained for 9,261 epochs and tested with $S = 10$ sampling steps on the RaindDrop dataset (Qian et al., 2018). The baseline RainDropDiff model results reported in the original paper of Özdenizci & Legenstein (2023) (which was trained for 37,042 epochs, 4 times more than our RestoreGrad models) are also presented here for comparison purposes.

| Model | $\eta$ | $\lambda$ | PSNR ↑ | SSIM ↑ |
|---|---|---|---|---|
| RestoreGrad (ours) | 0.05 | | **32.55** | **0.9440** |
| | 0.01 | | **32.73** | **0.9448** |
| | 0.005 | 0.1 | **32.69** | **0.9441** |
| | 0.001 | | **32.63** | 0.9404 |
| | 0.0005 | | **32.50** | 0.9405 |
| RestoreGrad (ours) | | 10 | **32.74** | **0.9442** |
| | 0.005 | 1 | **32.72** | **0.9441** |
| | | 0.1 | **32.69** | **0.9441** |
| | | 0.01 | **32.41** | 0.9417 |
| RainDropDiff (Özdenizci & Legenstein, 2023) | - | - | 32.29 | 0.9422 |

*Values in bold text indicate better scores than the baseline ReainDropDiff model.

Table 13: Comparison of baseline conditional DDPM (cDDPM) and the RestoreGrad on image super-resolution tasks.

| Methods | DIV2K ×2 | | | | DIV2K ×4 | | | |
|---|---|---|---|---|---|---|---|---|
| | PSNR ↑ | SSIM ↑ | LPIPS ↓ | FID ↓ | PSNR ↑ | SSIM ↑ | LPIPS ↓ | FID ↓ |
| Baseline cDDPM (Özdenizci & Legenstein, 2023) | 27.40 | 0.9291 | **0.127** | 7.577 | 25.18 | 0.8064 | **0.269** | 7.849 |
| + RestoreGrad (ours) | **27.56** | **0.9341** | 0.136 | **7.547** | **25.56** | **0.8228** | 0.290 | **7.839** |

*Better values are indicated with bold text.

**More image restoration examples.** We provide more examples in Figures 13, 15, 16 for comparing our RestoreGrad with the baseline DDPM approach (i.e., WeatherDiff) of Özdenizci & Legenstein (2023). Both models were trained on the multi-weather AllWeather dataset, where our RestoreGrad model was trained for only 887 epochs while WeatherDiff was trained for 1,775 epochs. The restored images of WeatherDiff were obtained by using the trained model weights provided by Özdenizci & Legenstein (2023) at `https://github.com/IGITUGraz/WeatherDiffusion`.

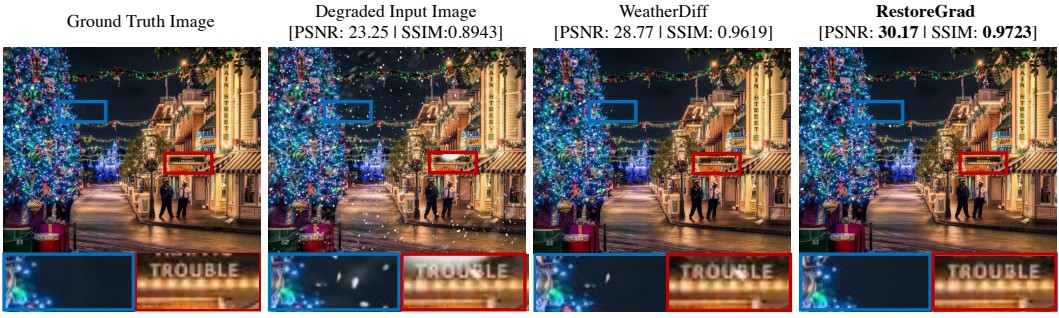

Figure 13: Image restoration examples using a test image taken from the Snow100K-L test set.

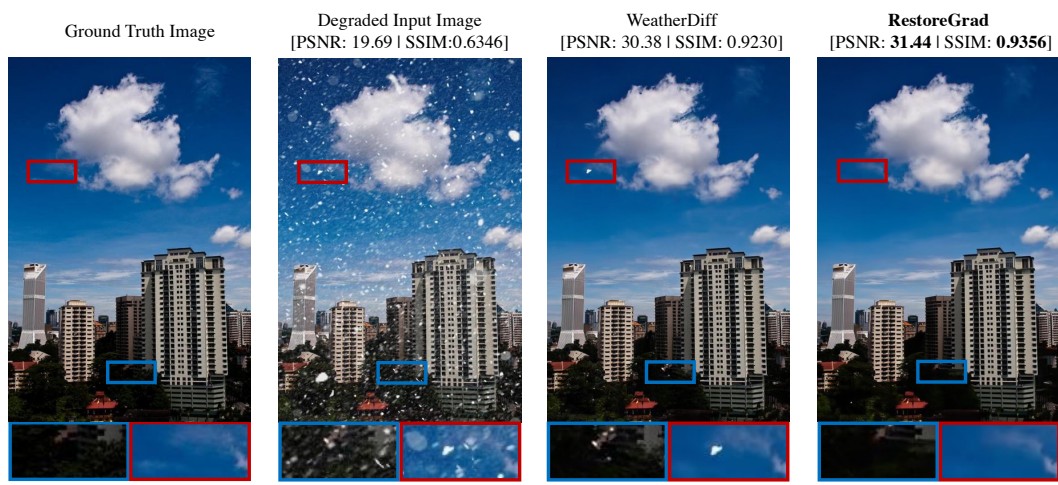

Figure 14: Image restoration examples using a test image taken from the Snow100K-L test set.

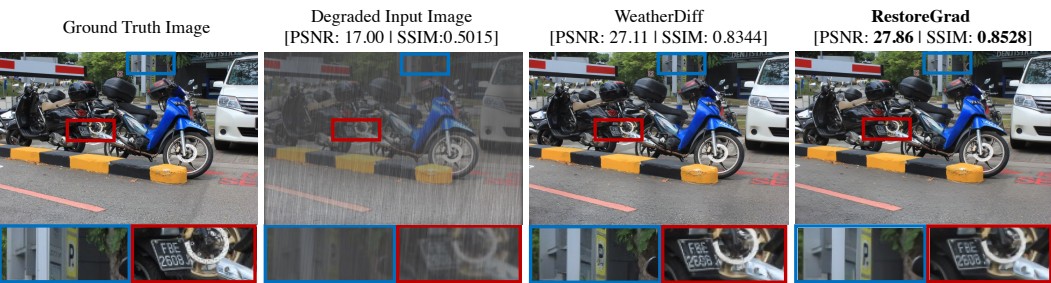

Figure 15: Image restoration examples using a test image taken from the Outdoor-Rain test set.

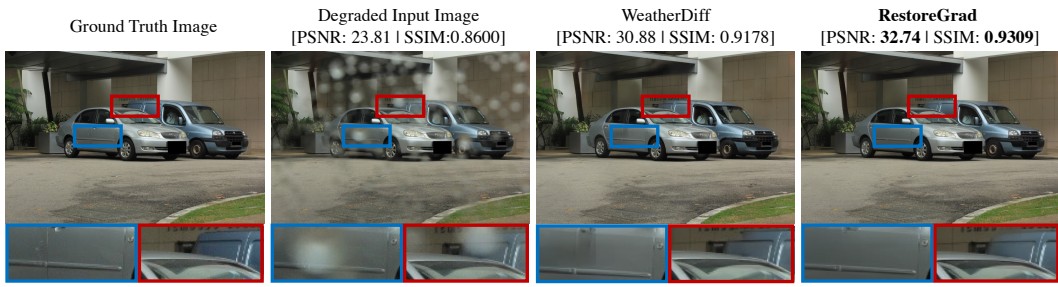

Figure 16: Image restoration examples using a test image taken from the RainDrop test set.

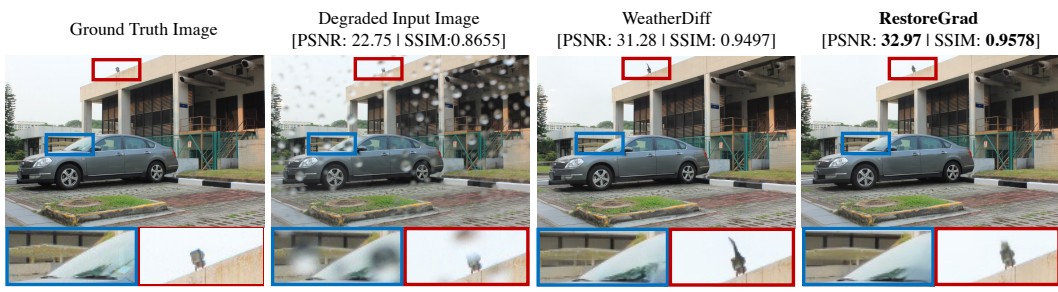

Figure 17: Image restoration examples using a test image taken from the RainDrop test set.

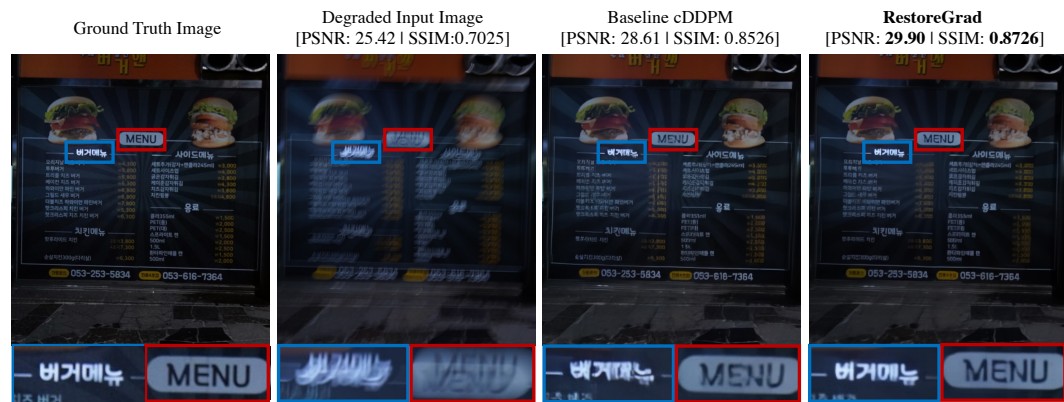

Figure 18: Image deblurring examples using a test image taken from the RealBlur test set.

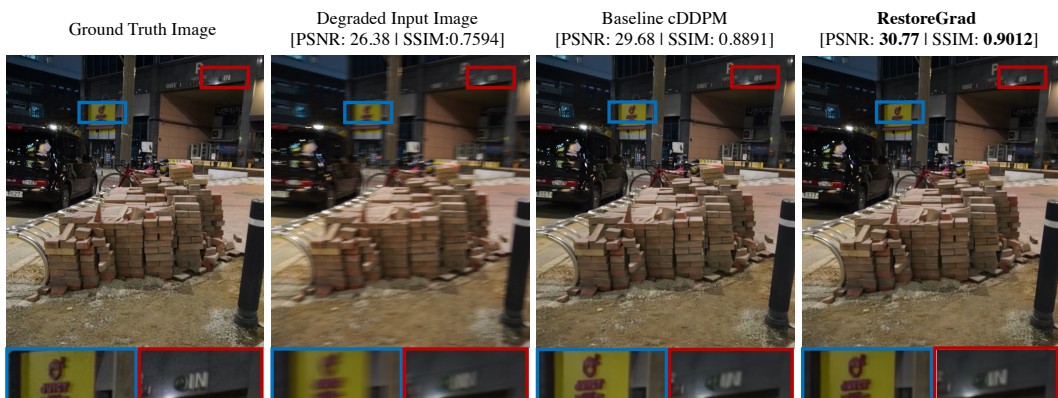

Figure 19: Image deblurring examples using a test image taken from the RealBlur test set.

