# OpenReview forum: "RestoreGrad: Signal Restoration Using Conditional Denoising Diffusion Models with Jointly Learned Prior"
_ICLR.cc/2025/Conference — Submitted to ICLR 2025_

### Official Review · Reviewer_VRxE · 2024-10-20

**Soundness:** 2
**Presentation:** 3
**Contribution:** 2
**Rating:** 5
**Confidence:** 3

**Summary:**

The paper proposes RestoreGrad, a method that aims to enhance conditional restoration DDPM models. The main idea is to add to the training 2 auxiliary networks that will train jointly with the original DDPM model and allow the DDPM to use at inference non-gaussian prior that is dependent on the degraded input. RestoreGrad is tested on image restoration and speech enhancement and shows faster convergence at training as well as robustness to inference with fewer steps. The quality of the results is either on-par or better for the reported metrics.

**Strengths:**

-	RestoreGrad converges faster with the same performances relative to the baselines, and the auxiliary nets do not add many parameters (only 2%).
-	The experiments contain comparisons against many SotA algorithms.
-	The paper is clear and well written.
-	The robustness to sampling using fewer timesteps is important and works well.

**Weaknesses:**

-	**Novelty:** A core claim of novelty is the idea of using a learned, non-gaussian prior. However, this idea is not new. For example, PriorGrad already discussed in the past this exact same idea (however not achieving better performances). The authors mention this and previous non-learened non-gaussian priors as different conceptually or technically (L88). However, Grad-TTS also used a learned prior, and successfully.
This limits the novelty contribution to only adaptation to the image domain.
-	**Evaluations:**
  -	The paper lacks subjective evaluations via user studies.
  -	In L1057 the authors claim higher PSNR yields better perceptual image quality, which is known to not be the case as it is a distortion metric only (See e.g., Blau & Michaeli 2018 or Freirich et al.2021).
  -	The important LPIPS and FID metrics are only present in the appendix for a single task (RainDropDiff). The authors should move them to the main part of the manuscript and add these metrics for the rest of the domains and models (and specifically in the tables and not only as graphs).
  -	There are no links to qualitative samples for the speech enhancement task.
-	**Minor**:
  -	In the abstract please clarify that you improve conditional DDPM for signal restoration (and not just general conditional models).
  -	L463 missing the device & train time (the latter is mentioned in later paragraph, but it's a bit unorganized).

**Questions:**

-	Will code be published on acceptance?
-	I would suggest maybe a more perceptual speech enhancement metric such as automatic speech recognition before and after the degradations, e.g. using whisper (see Benita et al. 2024).

---

> ### Author Response · Authors · 2024-11-22
> **Response to Reviewer VRxE (W1~W3)**
>
> ## Weaknesses
>
> > **W1:** A core claim of novelty is the idea of using a learned, non-gaussian prior. However, this idea is not new. For example, PriorGrad already discussed in the past this exact same idea (however not achieving better performances). The authors mention this and previous non-learened non-gaussian priors as different conceptually or technically (L88). However, Grad-TTS also used a learned prior, and successfully. This limits the novelty contribution to only adaptation to the image domain.
>
> We thank the reviewer for raising their concern about the novelty of RestoreGrad. We would like to provide further clarification to our contributions by pointing out the main differences with the mentioned prior works, PriorGrad [R1] and Grad-TTS [R2].
>
> **Differentiation from PriorGrad [R1]:**
> - Our proposed framework learns the prior distribution for conditional DDPMs by employing **two encoders**, i.e., Prior Net and Posterior Net, **to improve the learning of the prior by exploiting posterior information from both the original clean signal $\mathbf{x}_0$ and the conditioning signal $\mathbf{y}$.** This design was not utilized in PriorGrad nor in other works, but found to be beneficial for obtaining better performance in our work. We also invite the reviewer to check the supporting evidence for this design in Table 10 of the revised manuscript or related discussion at [Response to Reviewer Z1ga: W1](https://openreview.net/forum?id=UbMYhX60tY&noteId=TXEgdJgO5b).
> - Our framework successfully demonstrates the **feasibility** of **jointly** learnable prior ideas for improved conditional DDPM efficiency and performance, which PriorGrad failed in showing so.
>
> **Differentiation from Grad-TTS [R2]:**
> - Our framework aims at improving **general signal restoration applications** which learns the prior in conjuncture with the DDPM model by adopting a two-encoder design of Prior Net and Posterior Net, while Grad-TTS utilized a single encoder module and training scheme that were specifically designed for the text-to-speech synthesis task.
>
> - We formally derived our algorithm from the ELBO perspective under the DDPM framework, while Grad-TTS was developed in the context of score-based continuous-time diffusion models.
>
> To conclude, our method introduces a new prior learning framework adopting **two encoders** to effectively learn the diffusion prior **in conjuncture** with the DDPM model for signal restoration, improving both training and sampling efficiency and generalizable to various signal modalities. We believe its **effectiveness** and **generality** can provide significant contributions to diffusion-based models, and hopefully to motivate more interesting research in the future.
>
> - [R1] Lee et al., "PriorGrad: Improving conditional denoising diffusion models
> with data-dependent adaptive prior," ICLR, 2021.
>
> - [R2] Popov et al., "GradTTS: A diffusion probabilistic model for text-to-speech," ICML, 2021.
>
> > **W2: The paper lacks subjective evaluations via user studies.**
>
> We thank the reviewer for pointing out the weakness of lacking subjective evaluations through user studies. We totally agree with the reviewer that user studies will be helpful for improving validation of our method, but unfortunately, we have to admit that we were unable to conduct such evaluations in this limited time.
> In an effort to make up for the lack, we have provided audio samples via an anonymous link (https://anonymous.4open.science/r/SE_audio_samples-2D7C/) in the revised PDF for the SE task to further assist qualitative evaluation. We also provided ASR-based evaluation based on your constructive suggestion in the later comment. For the IR tasks, we hope the image examples provided in Appendix C.2 are helpful for facilitating perceptual evaluation.
>
> > **W3:** In L1057 the authors claim higher PSNR yields better perceptual image quality, which is known to not be the case as it is a distortion metric only (See e.g., Blau \& Michaeli 2018 or Freirich et al.2021).
>
> Thank you for pointing out this unsuitable description of the PSNR metric that we had in the original manuscript. We have made corresponding corrections in the revised manuscript where appropriate to avoid improper understanding of the PSNR metric as a perceptual quality metric and pointing to the two suggested papers [R3] (Blau \& Michaeli 2018) and [R4] (Freirich et al.2021) as reference.
>
> - [R3] Blau and Michaeli, "The perception-distortion tradeoff," CVPR, 2018.
>
> - [R4] Freirich et al, "A theory of the distortion-perception tradeoff in
> wasserstein space," NeurIPS, 2021.

---

> ### Author Response · Authors · 2024-11-22
> **Response to Reviewer VRxE (W4~W7, Q1~Q2)**
>
> > **W4:** The important LPIPS and FID metrics are only present in the appendix for a single task (RainDropDiff). The authors should move them to the main part of the manuscript and add these metrics for the rest of the domains and models (and specifically in the tables and not only as graphs).
>
> We appreciate the reviewer's suggestion on more leveraging the LPIPS and FID metrics for evaluating the image results. In the revised manuscript we have replaced the PSNR and SSIM plots in Figure 2 by the LPIPS and FID plots which were originally placed in the appendix section. We have also added LPIPS and FID numbers to Table 7 and Table 13 in the revised manuscript for the baseline cDDPM and our RestoreGrad, to make the comparison more complete. Note that the original works of the SOTA methods in Table 3 and Table 4 did not provide LPIPS and FID measures, so we did not further include them in the tables.
>
> > **W5:** There are no links to qualitative samples for the speech enhancement task.
>
> We prepared an anonymous link (https://anonymous.4open.science/r/SE_audio_samples-2D7C/) to audio samples for our speech enhancement task in the revised PDF.
>
> > **W6:** In the abstract please clarify that you improve conditional DDPM for signal restoration (and not just general conditional models).
>
> We are grateful for the reviewer's suggestion on the clarification and have modified the abstract accordingly in the revised manuscript to make it clear that we are improving conditional DDPMs for signal restoration applications.
>
> > **W7:** L463 missing the device \& train time (the latter is mentioned in later paragraph, but it's a bit unorganized).
>
> We thank the reviewer for bringing up the missing information in the main text of the original manuscript. We have added the device type and training time information to the experimental setup sections (Section 5.1.1 (SE) and Section 5.2.1 (IR)) of the revised manuscript.
>
> ## Questions
>
> > **Q1:** Will code be published on acceptance?
>
> Our organization has a rather strict policy on publicizing code, and as a result we are unfortunately unable to guarantee anything regarding sharing code to the public domain at the moment. However, as the experiments presented in this paper are performed based on using open-source code repositories and datasets, we have tried our best to provide the implementation details as sufficient as possible with associated website links in the manuscript. Still, it is possible that some puzzles have been missed. Thus, we are willing to provide more details required for reproduction through the email correspondence. Furthermore, we are checking with our legal team and trying to find out the maximal information that can be made public if our paper gets accepted.
>
> > **Q2:** I would suggest maybe a more perceptual speech enhancement metric such as automatic speech recognition before and after the degradations, e.g. using whisper (see Benita et al. 2024).
>
> We are grateful for the reviewer's suggestion on using the automatic speech recognition (ASR) as an additional metric to augment our results on speech enhancement. We feel this is very helpful for further validating our models. We have thus conducted ASR evaluation and added the new results to our revised manuscript (Appendix C.1, Table 10) which is also presented below for your ease of access.
>
> **Table:** *Following [R5] who perform evaluation of diffusion-based speech generation using ASR, we evaluate the SE model as a front-end denoiser for ASR under noisy environments. We test the diffusion-based SE models on two ASR engines (Conformer [R6], Citrinet [R7]) for the VoiceBand+DEMAND test set in terms of word-error-rate (WER) and character-error-rate (CER). It is interesting to see that CDiffuSE and PriorGrad actually lead to worse performance than the unprocessed speech case for Citrinet ASR, while only RestoreGrad improves the WER and CER and achieves the best results for both ASR models. The results further confirm the superiority of RestoreGrad over the baseline and PriorGrad for generating cleaner speech.*
>
> | SE model      | ASR: Conformer [R6]  | ASR: Citrinet [R7]   |
> |---------------|----------------------|----------------------|
> | Unprocessed   | 6.62 / 6.15          | 8.69 / 6.86          |
> | CDiffuSE      | 6.55 / 6.01          | 9.77 / 7.41          |
> | + PriorGrad   | 6.13 / 5.70          | 9.15 / 7.00          |
> | + RestoreGrad | **5.07** / **5.27** | **8.15** / **6.51** |
>
> *Best values are indicated with bold text. The numbers are presented in the format of WER / CER (\%).
>
> - [R5] Benita et al., "DiffAR: Denoising diffusion autoregressive model
> for raw speech waveform generation" ICLR, 2024.
>
> - [R6] Gulati et al., "Conformer: Convolution-augmented transformer for speech recognition" Interspeech, 2020.
>
> - [R7] Majumdar et al., "Citrinet: Closing the gap between non-autoregressive and autoregressive end-to-end models for automatic speech recognition," arXiv preprint arXiv:2104.01721, 2021.

---

> > ### Comment · Reviewer_VRxE · 2024-11-25
> >
> > Thanks for the detailed response. The samples are quite helpful and the new metrics as well. I still believe a user study is important here.
> > Additionally, while the qualitative samples and additional metrics help me agree more on the superiority of RestoreGrad, I remain unconvinced with regards to the innovation (which seems to agree with Reviewer WuzF).
> > I will therefore update my score to reflect that.

---

> ### Author Response · Authors · 2024-11-27
> **Official Comment by Authors**
>
> Thank you for your time and effort, we greatly appreciate your response. We are grateful that you have increased the score and become more positive on our work. It was encouraging. In addition, we are glad that you find the audio samples and new metrics helpful for better appreciating the superiority of our method. Before the end of the discussion period, we are also happy to provide further clarification to any specific questions regarding the novelty aspect where you might have concerns with. If there are further suggestions or comments that you think can help us improve the work further, please also do not hesitate to let us know and we will be excited to discuss. Thank you!

---

> ### Author Response · Authors · 2024-12-02
> **Official Comment by Authors before extended discussion period closes**
>
> We greatly appreciate the reviewer's constructive suggestions and comments that have helped us improve the paper considerably. We kindly remind the reviewer that there are only two days left before the closure of the discussion period, and we are happy to answer any further questions. We also sincerely invite the reviewer to look at our latest general response at ["Official Comment by Authors before End of Discussion Period"](https://openreview.net/forum?id=UbMYhX60tY&noteId=i2YnkJlbL7) if you have not had a chance, where we have provided more detailed and explicit comparison with existing approaches. We hope you find the discussion there helpful and interesting, and that it makes you more convinced about our innovation. Otherwise, we are happy to further address any specific concern during the remaining time for discussion. Once again, thank you for your valuable time and effort and we hope you have enjoyed reading our paper.

---

### Official Review · Reviewer_RjT2 · 2024-11-01

**Soundness:** 2
**Presentation:** 2
**Contribution:** 3
**Rating:** 6
**Confidence:** 3

**Summary:**

This paper proposes a method to improve the approach of using diffusion models conditioned on degraded data to solve restoration tasks in the audio and image domains. This method is inspired by PriorGrad, which uses a data-dependent adaptive prior, and attempts to extend this approach. Specifically, it simultaneously trains a Prior Net that estimates the parameters of the data-dependent prior. During training, it also simultaneously trains a Posterior Net that estimates the parameters of the data-dependent prior using the true data x_0 as input, along with the main score function. While PriorGrad handcrafts the prior, this method learns it simultaneously. The model is trained with a modified ELBO as the objective function, and experiments show that the proposed method converges 5 to 10 times faster in training compared to the baseline and achieves good performance with 2 to 2.5 times fewer steps.

**Strengths:**

- The experiments in Section 5 demonstrate that RestoreGrad shows notable improvements over baseline models in both speech and image restoration tasks. This method contributes significantly to the field as it has a certain level of generalizability, meaning it can be applied to many techniques with similar frameworks.
- The number of parameters in the Prior Net and Posterior Net is only about 2% of the main diffusion model's total parameters. This means that the additional computational/memory cost is kept to a minimum.
- This method includes hyperparameters \eta and \lambda that balance the weights in the loss function. However, the performance of the model doesn't seem to be very sensitive to the values of these parameters. This is a significant advantage for practitioners who will use this method in real-world applications.

**Weaknesses:**

- While one of the strength of this method lies in its generalizability, the speech-related experiments only use CDiffSE (proposed in 2022) as a baseline. This alone might not be sufficient to fully demonstrate the validity of the method. It seems necessary to address its applicability to other techniques as well.
- In the field of SE, there are clear improvements compared to the baseline. However, looking at Table 3, it appears that the improvements in the image processing domain are not as significant.

**Questions:**

1. In addition to PriorGrad, I understand that SpecGrad is also related to this paper and shares similar objectives, albeit for different tasks. Could you explain the relationship with SpecGrad and discuss the advantages of this method?
2. While this method seems applicable to various techniques, what are its limitations? For instance, does it require the amplitude information of the signal to be significant? If so, is it difficult to apply to latent diffusion model-based methods? This is briefly mentioned in the conclusion, but could you elaborate on this a bit more?
3. Can you provide more insights into the values output by the Prior Net (or the Posterior Net output)? How does the characteristics of the output signal sequence differ from PriorGrad or SpecGrad? I felt it was outputting the amplitude information of the signal to be estimated in the restoration task. If so, is it possible to replace this prior net module with the output of a fixed SE module?
4. Comparing the SE results with the IR results, the improvements in SE seem more pronounced. Can you discuss this difference? Is it possible to discuss this in light of the characteristics of each type of data? (For example, is it important that speech has a zero mean?)

---

> ### Author Response · Authors · 2024-11-22
> **Response to Reviewer RjT2 (W1~W2, Q1)**
>
> ## Weaknesses
>
> > **W1:** While one of the strength of this method lies in its generalizability, the speech-related experiments only use CDiffSE (proposed in 2022) as a baseline. This alone might not be sufficient to fully demonstrate the validity of the method. It seems necessary to address its applicability to other techniques as well.
>
> We thank the reviewer for acknowledging the generalizability of our method as a strength, and for raising your concern on the speech enhancement (SE) baseline model considered in this work, i.e., CDiffuSE [R1] (Lu et al., ICASSP 2022). We explain the reason of using CDiffuSE as the baseline and also update our SOTA comparison.
>
> **Why we chose CDiffuSE as baseline:** As one of the pioneering DDPM-based approaches for SE, CDiffuSE is undoubtedly a representative diffusion-based SE model that has gained much attention and motivated later models. It is worth noting that, CDiffuSE builds upon the DiffWave model [R2] (Kong et al., ICLR 2021), which is known for its versatility for various audio generation tasks.
> Showing the benefits of our joint prior learning idea on CDiffuSE implies the potential to also improve DiffWave-based methods, thereby the applicability to other audio applications and techniques.
>
> **Comparison to stronger SOTA model:** We have included a more recent SOTA model, DOSE [R3] (Tai et al., NeurIPS 2023), which is a diffusion-based SE model also built upon the DiffWave architecture.
> As shown in the revised manuscript, our method performs slightly inferior to DOSE for the in-domain VoiceBand+DEMAND data (Table 2), while better on out-of-domain CHiME-3 data (Table 6), notably achieved by using only $4.6\times$ fewer training iterations then DOSE. We hope adding this stronger baseline can improve the completeness of our comparison.
>
> - [R1] Lu et al., "Conditional diffusion probabilistic model for speech enhancement," ICASSP, 2022.
>
>
> - [R2] Kong et al., "DiffWave: A versatile diffusion model for audio synthesis," ICLR, 2021.
>
> - [R3] Tai et al., "DOSE: Diffusion dropout with adaptive prior for speech enhancement," NeurIPS, 2023.
>
> > **W2:** In the field of SE, there are clear improvements compared to the baseline. However, looking at Table 3, it appears that the improvements in the image processing domain are not as significant.
>
> We appreciate that the reviewer acknowledged our clear improvements in the speech domain. We would also like to argue that our improvements in the image domain are equally significant. The main reason is the baseline IR model we consider in this paper, i.e., the patch-based denoising diffusion model [R4] (Ozdenizci \& Legenstein, IEEE TPAMI 2023), is already a powerful model achieving SOTA performance on various IR benchmarks. Therefore, **improving upon this baseline IR model is already challenging, and being able to achieve the improvements with only minimum increase of complexity, is worth being considered as significant**. Second, in addition to the IR tasks on adverse weather conditions that [R4] originally considered, we further demonstrate our method's merits for other types of IR tasks (image debluring and super-resolution). The image quality improvements on various degradation types can also be clearly observed and well perceived through the image examples provided in Figure 6 and Figures 13-19 in the revised PDF. **We deeply feel such improvements upon the rather strong baseline and across various restoration tasks stand for the significance of our method in the image domain.**
>
> [R4] Ozdenizci \& Legenstein, "Restoring vision in adverse weather conditions with patch-based denoising diffusion models," IEEE TPAMI 2023.
>
> ## Questions
>
> > **Q1:** In addition to PriorGrad, I understand that SpecGrad is also related to this paper and shares similar objectives, albeit for different tasks. Could you explain the relationship with SpecGrad and discuss the advantages of this method?
>
> Yes, definitely. SpecGrad is a follow-up work of PriorGrad based on using an adaptive prior for neural vocoders as mentioned in the Related Work section. To provide more details here, SpecGrad utilizes spectral information of the conditioning signal to form the covariance matrix $\boldsymbol{\Sigma}$ of the diffusion prior that leads to closer distribution of the target signal, thereby improving the performance over PriorGrad. To achieve improvements, SpecGrad utilizes specifically designed spectral filters and also a full covariance matrix rather then a diagonal one, which in the same time could lead to increased complexity in sampling. In contrast, our RestoreGrad automatically learns a beneficial prior from the data rather than having to handcraft or design such spectral filters specifically used in speech processing, thus is more generic and applicable to other signal modalities and tasks.

---

> ### Author Response · Authors · 2024-11-22
> **Response to Reviewer RjT2 (Q2~Q4)**
>
> > **Q2:** While this method seems applicable to various techniques, what are its limitations? For instance, does it require the amplitude information of the signal to be significant? If so, is it difficult to apply to latent diffusion model-based methods? This is briefly mentioned in the conclusion, but could you elaborate on this a bit more?
>
> Yes, for sure. As the reviewer mentioned, we describe the main limitation of our method as currently focusing on signal restoration applications. To elaborate more on it, we note that our approach depends on the decent correlation between the target data (clean) and the conditioning data (noisy). In typical signal restoration problems, as the noisy signal usually preserves the structure of the target signal (e.g., in the amplitude domain) to some extent, they naturally correlate with each other to a certain level, and our method takes advantage of that. However, exploiting correlation in a more implicit domain, such as in the hidden space of a latent diffusion model (LDM), could still be possible while may also require more effort on studying and analyzing the characteristics of the signals in the hidden representation space. It might also require properly designing specialized architectures or even loss functions for different encoders/decoders of LDMs. Our current work has not addressed this aspect, while it will be very interesting to explore related research questions in the future.
>
> > **Q3:** Can you provide more insights into the values output by the Prior Net (or the Posterior Net output)? How does the characteristics of the output signal sequence differ from PriorGrad or SpecGrad? I felt it was outputting the amplitude information of the signal to be estimated in the restoration task. If so, is it possible to replace this prior net module with the output of a fixed SE module?
>
> In the speech enhancement (SE) task, one can observe that the output of Prior Net  follows the shape of the target signal's amplitude (e.g., see Figure 3).
> This observation makes sense, as by approximating the prior to the clean signal distribution, the diffusion process trajectory can be much improved as starting from somewhere closer to the distribution of the target. Notably, our framework exploits the clean speech $\mathbf{x}_0$ information during training as input to the Posterior Net, thereby enabling the Prior Net to learn a distribution that is closer to the clean speech signals for improved performance (for example, see our new results of Table 10 in the revised manuscript or related discussion at [Response to Reviewer Z1ga: W1](https://openreview.net/forum?id=UbMYhX60tY&noteId=TXEgdJgO5b)). On the contrary, when applying PriorGrad or SpecGrad to the SE task, the prior will be computed solely based on the noisy signal $\mathbf{y}$ and thus can become noisier, leading to reduced effectiveness.
>
> **How about replacing Prior Net by SE module?** Given the fact that RestoreGrad takes advantage of the closer-to-target prior for improvements, it might be possible to replace the Prior Net with a pre-trained SE model which outputs cleaner speech. However, by doing so it could also lead to increased complexity -- the SE model typically has a larger size (millions of parameters) than the Prior Net which is only around 100K parameters. In addition, pre-training of the SE model might be required for achieving good performance, which also results in increased training time. Therefore, RestoreGrad serves as a more suitable solution when model complexity and training efficiency are of important considerations.
>
> > **Q4:** Comparing the SE results with the IR results, the improvements in SE seem more pronounced. Can you discuss this difference? Is it possible to discuss this in light of the characteristics of each type of data? (For example, is it important that speech has a zero mean?)
>
> Since the SE and IR tasks are different in nature, and the evaluation metrics used for speech also differ from those for images considerably, we actually feel it is somewhat not so apparent whose improvements are more pronounced than the other. But one thing we could point out is that the DDPM model in the IR tasks is much larger than the prior and posterior encoder modules used for RestoreGrad. Consequently, the DDPM itself might be already powerful, and considerably improving upon it could be more challenging given the relatively much smaller encoder modules. We also do not think the model performance is affected by the characteristics of speech or images in our studies, as we performed normalization to both data before inputting them to the model. We feel that the performance is more correlated to the model capacities or hyperparameter settings rather than the type of data (for example, see our additional results of Table 11 in the revised PDF or related discussion at [Response to Reviewer WuzF: W7](https://openreview.net/forum?id=UbMYhX60tY&noteId=afzQMeVI8n)).

---

> > ### Comment · Reviewer_RjT2 · 2024-11-26
> >
> > Thank you for your thorough response to the rebuttal. I feel that the concerns I raised have been largely addressed. While I acknowledge the generality of this framework for a wide range of restoration tasks, I believe there are still points that require consideration: 1. The difficulty in applying this method to LDM-based approaches. 2. The possibility that the benefits provided by the Posterior Net could potentially be replaced by other regression-based networks (although I recognize the advantages in terms of training speed and the lightweight network structure). However, these considerations likely go beyond the scope of this paper and the current reviewing phase. Nevertheless, I consider the contribution to the field to be significant, and I have updated my score accordingly.

---

> ### Author Response · Authors · 2024-11-27
> **Official Comment by Authors**
>
> We sincerely thank you for the time and effort. We appreciate that you have increased the score and acknowledge the significance of RestoreGrad's contributions to the field. It was very encouraging. Meanwhile, we are happy to see that our responses have addressed your major concerns during the rebuttal. We also appreciate the discussion with you on potential expansions of our framework, which we will definitely like to explore in the future. If you have any further questions or comments that you think might be helpful for improving our work, please feel free to ask and we are happy to discuss.

---

> ### Author Response · Authors · 2024-12-02
> **Official Comment by Authors before extended discussion period closes**
>
> We greatly appreciate the reviewer's insightful feedback and suggestions that have helped us improve the paper a lot. We kindly remind the reviewer that there are only two days left before the closure of the discussion period, and we are happy to answer any further questions or comments that might come up during this short period. We also sincerely invite the reviewer to look at our latest general response at ["Official Comment by Authors before End of Discussion Period"](https://openreview.net/forum?id=UbMYhX60tY&noteId=i2YnkJlbL7) if you have not had a chance, which we hope you find interesting and helpful and makes you more supportive of our paper. Once again, thank you for your valuable time and effort and your positive view on our work.

---

### Official Review · Reviewer_Z1ga · 2024-11-03

**Soundness:** 3
**Presentation:** 3
**Contribution:** 3
**Rating:** 6
**Confidence:** 4

**Summary:**

This paper focuses on signal restoration tasks and explores diffusion models with learned prior distributions capturing information about corrupted signal $y$. The pipeline the authors propose resembles VAE with the decoder represented by a conditional diffusion model and two encoders. Thus, at inference information about corrupted signal is used not only as conditioning to the network estimating diffusion noise, but also parameterizes the diffusion prior. Three networks are trained jointly, and the posterior encoder (skipped at inference) takes as input both clean signal $x_0$ and its corrputed verison $y$ and helps the prior encoder learn data-dependent task-specific prior. The novel loss function providing upper bound on negative log-likelihood for such combined VAE-diffusion model is proposed. Experiments on speech enhancement and image restoration tasks show the benefits of the proposed approach compared to conditional diffusion models with data-independent priors or the ones based on heuristics when applicable.

**Strengths:**

1. The authors introduced a novel model design combining ideas from VAE and diffusion modeling: the proposed model consists of two encoders parameterising prior distribution and a conditional diffusion based decoder.
2. A novel ELBO-like loss was proposed allowing for stable training of the mentioned model. The learned priors look quite reasonable. Thus, this approach can become a good alternative to hand-crafted diffusion priors in signal restoration tasks.
3. The experiments clearly demonstrate that diffusion models can benefit from the priors learned in the way the authors propose. The number of training steps necessary to achieve the same performance level decreases compared to data-independent or hand-crafted priors. Moreover, RestoreGrad achieves better performance at convergence.
4. The presentation quality is good (except for one point, see below); the text is clear and the paper is easy to follow.

Overall, I think that this is a good paper from the point of view of originality and clarity.

**Weaknesses:**

I can point out three main weaknesses.

1. As I far as I understand, the role of the posterior net is to better capture the correlation between $x_0$ and $y$ and to pass this knowledge to the prior net whose only input is the corrupted signal $y$ which is achieved by prior matching terms in Formula (9). But what happens if we get rid of this posterior encoder? It is possible to think of the following design: both at training and inference we have the same scheme, the prior net $\psi$ is conditioned only on $y$ and parameterizes diffusion prior, and the conditional diffusion $\theta$ restores a signal from this prior. This model wouldn't differ from the one the authors propose at inference. As for training loss, we obviously would not have prior matching terms anymore, and  $\Sigma_{post}$ would be replaced with $\Sigma_{prior}$ in the latent regularization and denoising matching terms. Such a design looks more simple (only one encoder instead of two) and also allows for learnable prior. I think that it could be a good baseline in the experimental section and it could help to support the idea that the posterior encoder is helpful since it is trained on both $y$ and $x_0$.

2. Non-diffusion baselines in the experimental section seem to be a bit outdated. E.g., in speech enhancement best methods have started to achieve > 3.0 PESQ on a common SE benchmark (Valentini dataset) since the year 2022 [1] outperforming CDiffuSE and RestoreGrad, and you just mention quite old methods with PESQ < 2.5. Although I don't have experience with image deblurring, I also suspect that the baselines you choose for this task are far from SOTA because the papers you refer to were published no later than in 2020.

3. What is $\sigma_t$ in Formula (9)? On the one hand, in the text of the paper $\sigma_t$ is mentioned in the context of common diffusion models (e.g. lines 153 or 189) and it suggests that the reverse process $p_{\theta}$ has fixed (unlearnable) covariance matrix $\Sigma_{\theta}=\sigma^2_t I$. In your setting it must mean something different, since the Algorithm 2 in the Appendix suggests that at each reverse diffusion step the noise you add has covariance matrix $\sigma^2_t\Sigma_{prior}$. Anyway, you should define the reverse process $p_{\theta}$ in your setting more clear, like in lines 188-190, especially since you use $\sigma_t$ that seems to have something to do with it in Proposition 1.

Also, I feel there's something wrong with the values of $\gamma_t$ in Proposition 1. In practice you set them all to $1$, so it is not a big issue, but still I'll mention my concerns. On the one hand, in lines 250-251 and 824-825 it is defined as $\beta_t^2/2\sigma_t^2\alpha_t(1-\bar{\alpha_t})$ for $t>1$. On the other hand, in the proof of Proposition 1 we have lines 858-859 implying a different expression $\gamma_{t}=\beta_t/2\alpha_t(1-\bar{\alpha}_{t-1})$.

[1] Dual-branch attention-in-attention transformer for single-channel speech enhancement, Yu et al.

**Questions:**

1. I have a question regarding the proof in lines 858-860: we see that the expectation of KL divergence between two Gaussians $q(x_{t-1}|x_t,x_0)$ and $p_\theta(x_{t-1}|x_t,y)$ contains only MSE term $\Vert\epsilon-\epsilon_{\theta}\Vert_{\Sigma^{-1}}^2$ and does not contain the log-determinant terms implying that it is KL divergence between two Gaussians with the same covariance matrices $\Sigma_A$ and $\Sigma_B$ (and $\log\frac{\Sigma_A}{\Sigma_B}$ cancels out).  In the assumptions in Proposition 1 you explicitly define how the forward diffusion paths look like. It's easy to see that for such forward diffusions the covariance of $q(x_{t-1}|x_t,x_0)$ is $\tilde{\beta}_{t}\Sigma$. So, is it true that reverse process has this covariance matrix?

2. One of latent regularization terms is multiplied by $\alpha_T$ which should be very little since the mean of the prior should be very close to zero. So, this term should have very little impact in theory. Why do you still use it at training? Does it have any impact on the training in practice?

3. Is the training process sensitive to the choice of loss weights $\lambda$ and $\eta$?

Also, I'd like to make a couple of additional remarks here. These are just the remarks to consider, not questions:

4. Typo in line 043: "DDPMs typically consistS..."

5. It's interesting to see what happens with your model if we consider continuous-time diffusion formalism as in [2]. Continuous-time diffusions also allow for likelihood estimation (but in a different way, without ELBO-like derivation). In my understanding, in this case prior matching terms in the total loss (9) would remain unchanged, denoising matching terms would look similar, but I'm not sure about Latent Regularization terms. Would they still be there in the loss function? If they would, what whould the exact expression look like, and how would these changes affect model training/final performance?

[2] Score-Based Generative Modeling through Stochastic Differential Equations, Song et al.

---

> ### Author Response · Authors · 2024-11-22
> **Response to Reviewer Z1ga (W1)**
>
> ## Weaknesses
>
> > **W1:** As I far as I understand, the role of the posterior net is to better capture the correlation between $\\mathbf{x}\_0$ and $\\mathbf{y}$ to pass this knowledge to the prior net whose only input is the corrupted signal $\\mathbf{y}$ which is achieved by prior matching terms in Formula (9). But what happens if we get rid of this posterior encoder? It is possible to think of the following design: both at training and inference we have the same scheme, the prior net $\\psi$ is conditioned only on $\\mathbf{y}$ and parameterizes diffusion prior, and the conditional diffusion $\\theta$ restores a signal from this prior. This model wouldn't differ from the one the authors propose at inference. As for training loss, we obviously would not have prior matching terms anymore, and $\\boldsymbol{\\Sigma}\_{\\text{post}}$ would be replaced with $\\boldsymbol{\\Sigma}\_{\\text{prior}}$ in the latent regularization and denoising matching terms. Such a design looks more simple (only one encoder instead of two) and also allows for learnable prior. I think that it could be a good baseline in the experimental section and it could help to support the idea that the posterior encoder is helpful since it is trained on both $\\mathbf{y}$ and $\\mathbf{x}\_0$.
>
> We greatly appreciate the reviewer for suggesting such an interesting idea to check a potentially simpler variant of our learnable prior framework -- using only a single encoder Prior Net $\\psi$ for both training and inference. We have further conducted the experiments, where we trained the RestoreGrad model on the following loss function:
>
> $
> \\min\_{\\theta,\\psi} \\,\\,\\, \\eta\\bigr(\\underbrace{\\bar{\\alpha}\_T||{\\mathbf{x}\_0}||^2\_{\\boldsymbol{\\Sigma}^{-1}\_{\\text{prior}}}+\\log|{\\boldsymbol{\\Sigma}\_{\\text{prior}}|}}\_{\\mathcal{L}\_{\\text{LR}}}\\bigr)+\\underbrace{||{\\boldsymbol{\\epsilon}-\\boldsymbol{\\epsilon}_\\theta(\\mathbf{x}\_t,\\mathbf{y},t)}||^2\_{\\boldsymbol{\\Sigma}^{-1}\_{\\text{prior}}}}\_{\\mathcal{L}\_{\\text{DM}}},
> $
>
> which basically removes the Posterior Net $\\phi$ and only trains the Prior Net $\\psi$ and DDPM $\\theta$. In this case, both training and testing become the same scheme. We compare the models trained with the above equation for a few $\eta$ values with the baseline CDiffuSE, PriorGrad, and RestoreGrad models for the SE task in the table below. All models were trained with 96 epochs, inferred with 6 steps. From the results we observe that i) the learnable prior scheme without Posterior Net also works to some extent, outperforming the baseline CDiffuSE with suitable $\\eta$ values; ii) **RestoreGrad achieves better results with Posterior Net than without it**, indicating it benefits from being informed of the target signal $\\mathbf{x}\_0$ through the posterior encoder during training. We thank the reviewer again for suggesting this experiment, which has definitely helped us provide more insight to our developed framework. We have also included relevant discussion in the revised PDF (Section 5.1.2, Table 3).
>
> **Table:** *SE comparison of RestoreGrad models trained with and without Posterior Net. The better results with Posterior Net than without it indicate that the posterior encoder is helpful by exploiting the information of clean signal $\\mathbf{x}\_0$ during training.*
>
> | SE model                                      | PESQ $\uparrow$ | COVL $\uparrow$ | SSNR $\uparrow$ | SI-SNR $\uparrow$ |
> |-----------------------------------------------|-----------------|-----------------|-----------------|-------------------|
> | CDiffuSE (96 epochs)                          | 2.32            | 2.89            | 3.94            | 11.84             |
> | + PriorGrad                                   | 2.42            | 3.03            | 5.53            | 14.21             |
> | + RestoreGrad                                 | **2.51**   | **3.14**   | **5.92**   | **14.74**    |
> | + RestoreGrad w/o Posterior Net ($\eta=0$)    |   ------       | training        | diverged        | --------             |
> | + RestoreGrad w/o Posterior Net ($\eta=0.01$) | 2.47            | 3.08            | 4.96            | 11.22             |
> | + RestoreGrad w/o Posterior Net ($\eta=1$)    | 2.48            | 3.12            | 5.11            | 13.29             |
>
> *Best values are indicated with bold text.

---

> ### Author Response · Authors · 2024-11-22
> **Response to Reviewer Z1ga (W2~W3)**
>
> > **W2:** Non-diffusion baselines in the experimental section seem to be a bit outdated. E.g., in speech enhancement best methods have started to achieve > 3.0 PESQ on a common SE benchmark (Valentini dataset) since the year 2022 [1] outperforming CDiffuSE and RestoreGrad, and you just mention quite old methods with PESQ < 2.5. Although I don't have experience with image deblurring, I also suspect that the baselines you choose for this task are far from SOTA because the papers you refer to were published no later than in 2020.
>
> We thank the reviewer for raising the concern on the SOTA baselines. For the SE performance on the VoiceBand+DEMAND dataset (i.e., Valentini dataset), we mainly compare the performance with generative SE models, which usually lag behind the performance of the deterministic counterparts for in-domain data. However, generative SE models have demonstrated more robust performance to domain shift in the input data than deterministic SE models, owing to fitting the distribution of the training samples instead of optimizing a point-wise distance loss [R1]. Our primary goal in this work is to show that our framework can improve the conditional DDPM-based generative SE models, thus we mainly compare with generative SE methods given this context. To alleviate the concern on relatively outdated baseline SE, we have included a more recent diffusion-based SE model, DOSE [R2] (Tai et al., NeurIPS 2023), to Table 2 and Table 6 of the revised PDF. We found that our method performs comparably to the stronger SOTA model of DOSE for in-domain data (Table 2), and slightly better for out-of-domain data (Table 6), suggesting the effectiveness of our model to improve upon baseline CDiffuSE. We hope this can mitigate the reviewer's concern on the SOTA comparison.
>
> For the image-deblurring task, our main purpose is to demonstrate the improvements brought by RestoreGrad are applicable to degradation types other than the main weather degradation datasets, rather than beating the latest SOTA methods on deblurring tasks. Here, the baseline cDDPM used is not specifically designed for deblurring; it is just the same model used in the main experiments of weather degradations. Therefore, the performance of the model is not optimized for the deblurring task. The two presented existing approaches in Table 7 achieved similar performance as the baseline cDDPM, so we used them as the reference to quantify the improvement provided by RestoreGrad over the baseline cDDPM. We have noted this reason for choosing the two models in the revised manuscript.
>
> - [R1] Lu et al., "Conditional diffusion probabilistic model for speech enhancement," ICASSP, 2022.
>
> - [R2] Tai et al., "DOSE: Diffusion dropout with adaptive prior for speech enhancement," NeurIPS, 2023.
>
> > **W3:** What is $\\sigma\_t$ in Formula (9)? On the one hand, in the text of the paper $\\sigma\_t$ is mentioned in the context of common diffusion models (e.g. lines 153 or 189) and it suggests that the reverse process $p\_\\theta$ has fixed (unlearnable) covariance matrix $\\boldsymbol{\\Sigma}\_\\theta=\\sigma\_t^2\\mathbf{I}$. In your setting it must mean something different, since the Algorithm 2 in the Appendix suggests that at each reverse diffusion step the noise you add has covariance matrix $\\sigma\_t^2\\boldsymbol{\\Sigma}\_{\\text{prior}}$. Anyway, you should define the reverse process $p\_\\theta$ in your setting more clear, like in lines 188-190, especially since you use $\\sigma\_t$ that seems to have something to do with it in Proposition 1.
>
> We appreciate the reviewer for asking to clarify the variable $\\sigma\_t$ which was not clear in the original manuscript. Yes, as mentioned in lines 153 of the old manuscript, $\\sigma\_t^2=\\tilde{\\beta}\_t\\coloneq\\frac{1-\\bar{\\alpha}\_{t-1}}{1-\\bar{\\alpha\_t}}\\beta\_t$ according to eq. (4). In the standard DDPM, this comes from assuming the covariance of the reverse process $p\_\\theta$ to be the same as the true forward process posterior conditioned on $\\mathbf{x}\_0$, i.e., $\\boldsymbol{\\Sigma}\_\\theta(\\mathbf{x}\_t,t)=\\sigma\_t^2\\mathbf{I}$, where $\\sigma\_t^2=\\tilde{\\beta}\_t$. In our case, we also adopt this assumption to use the same covariance matrix of the true forward process posterior conditioned on $\\mathbf{x}\_0$ for the reverse process, while now the covariance becomes $\\sigma\_t^2\\boldsymbol{\\Sigma}\_{\\text{post}}$ as the noise $\\boldsymbol{\\epsilon}$ is sampled from $\\mathcal{N}(\\mathbf{0},\\boldsymbol{\\Sigma}\_{\\text{post}})$ during training in the forward process. Note that the definition of $\\sigma\_t$ is the same as in standard DDPM. We have modified Proposition 1 to make it more clear the assumption used for the covariance of the reverse process as well as the definition of $\\sigma\_t$ in the revised manuscript.

---

> ### Author Response · Authors · 2024-11-22
> **Response to Reviewer Z1ga (Q1~Q3)**
>
> > Also, I feel there's something wrong with the values of $\\gamma\_t$ in Proposition 1. In practice you set them all to 1, so it is not a big issue, but still I'll mention my concerns. On the one hand, in lines 250-251 and 824-825 it is defined as $\\beta\_t^2/2\\sigma\_t^2\\alpha\_t(1-\\bar{\\alpha}\_t)$ for $t>1$. On the other hand, in the proof of Proposition 1 we have lines 858-859 implying a different expression $\\gamma\_t=\\beta\_t/2\\alpha\_t(1-\\bar{\\alpha}\_{t-1})$.
>
> Both expressions are actually equivalent. To see this, note that we have $\\sigma\_t^2=\\tilde{\\beta}\_t\\coloneqq\\frac{1-\\bar{\\alpha}\_{t-1}}{1-\\bar{\\alpha\_t}}\\beta\_t$ by eq. (4) and line 153 of the original manuscript. Using this equation we can see that $\\gamma\_t=\\beta\_t^2/2\\sigma\_t^2\\alpha\_t(1-\\bar{\\alpha}\_t)=\\beta\_t/2\\alpha\_t(1-\\bar{\\alpha}\_{t-1})$ for $t>1$. We thank the reviewer for bringing up this question and we have made this expression more explicit in the proof of Proposition 1 of the revised manuscript.
>
> ## Questions
>
> > **Q1:** I have a question regarding the proof in lines 858-860: we see that the expectation of KL divergence between two Gaussians $q(\mathbf{x}\_{t-1}|\mathbf{x}\_t,\mathbf{x}\_0)$ and $p\_\theta(\mathbf{x}\_{t-1}|\mathbf{x}\_t,y)$ contains only MSE term $||\\boldsymbol{\\epsilon}-\\boldsymbol{\\epsilon}\_\\theta||^2\_{\\boldsymbol{\\Sigma}^{-1}}$ and does not contain the log-determinant terms implying that it is KL divergence between two Gaussians with the same covariance matrices $\\boldsymbol{\\Sigma}\_A$ and $\\boldsymbol{\\Sigma}\_B$ (and $\\log\\frac{\\boldsymbol{\\Sigma}\_A}{\\boldsymbol{\\Sigma}\_B}$ cancels out). In the assumptions in Proposition 1 you explicitly define how the forward diffusion paths look like. It's easy to see that for such forward diffusions the covariance of $q(\mathbf{x}\_t|\mathbf{x}\_t,\mathbf{x}\_0)$ is $\\tilde{\\beta}\_t\\boldsymbol{\\Sigma}$. So, is it true that reverse process has this covariance matrix?
>
> Yes, the covariance of the reverse process is assumed to match the true posterior conditioned on $\mathbf{x}\_0$ of the forward process, following the common practice in the DDPM literature (e.g., see [R3][R4]). In this case, the diffusion model $\theta$ only estimates the reverse process mean to predict the forward process mean $\tilde{\boldsymbol{\mu}}\_t(\mathbf{x}\_t,\mathbf{x}\_0)$ (which is actually realized by estimating the real noise $\boldsymbol{\epsilon}$ for better performance in DDPM [R5]). Although it is possible to also estimate the covariance using the neural network, it will lead to increased complexity while not necessarily improving the performance as reported in [R5]. We also follow the literature to set the reverse process covariance matrix to be the same as the forward process (or more exactly, the true forward process posterior conditioned on $\mathbf{x}\_0$).
>
> - [R3] Calvin Luo, "Understanding diffusion models: A unified perspective," arXiv preprint
> arXiv:2208.11970, 2022.
>
> - [R4] Ozdenizci \& Legenstein, "Restoring vision in adverse weather conditions with patch-based denoising diffusion models," IEEE TPAMI, 2023.
>
> - [R5] Ho et al., "Denoising diffusion probabilistic models," NeurIPS, 2020.
>
> > **Q2:** One of latent regularization terms is multiplied by
> $\alpha\_T$ which should be very little since the mean of the prior should be very close to zero. So, this term should have very little impact in theory. Why do you still use it at training? Does it have any impact on the training in practice?
>
> This is a good subtle question! Although $\alpha\_T$ is very small, the weighted norm term multiplied with it also has the inverse covariance matrix $\boldsymbol{\Sigma}\_{\text{post}}^{-1}$. Therefore, the value of the weighted norm could become very large in magnitude if the covariance matrix becomes close to singular. In that scenario, when the term is multiplied with $\alpha\_T$ it might not be negligible. In practice, we also found the performance degrades without having the term for training, and the model could become more sensitive to the loss weight $\eta$ (a possible reason is due to lacking counterbalance of the $\log|\boldsymbol{\Sigma}\_{\text{post}}|$ term if we do not have this particular term).
>
>
> > **Q3:** Is the training process sensitive to the choice of loss weights $\lambda$ and $\eta$?
>
> One of the advantages of our method is that we empirically found it not so sensitive to the choice of the loss weights, as demonstrated in Figure 10 and Table 12 in the revised manuscript. From there, we see that for a wide range of the loss weights RestoreGrad outperforms the baseline models.

---

> ### Author Response · Authors · 2024-11-22
> **Response to Reviewer Z1ga (Q4~Q5)**
>
> > **Q4 (remark):** Typo in line 043: "DDPMs typically consistS..."
>
> We thank the reviewer for pointing out the typo and have corrected it in the revised manuscript.
>
> > **Q5 (remark):** It's interesting to see what happens with your model if we consider continuous-time diffusion formalism as in [2]. Continuous-time diffusions also allow for likelihood estimation (but in a different way, without ELBO-like derivation). In my understanding, in this case prior matching terms in the total loss (9) would remain unchanged, denoising matching terms would look similar, but I'm not sure about Latent Regularization terms. Would they still be there in the loss function? If they would, what would the exact expression look like, and how would these changes affect model training/final performance?
>
> We greatly appreciate the reviewer's remark on extending our idea to the score-based diffusion models. This is a very interesting research direction that could lead to impactful contributions, since continuous-time diffusion models are another popular class of generative models adopted in many applications besides DDPMs. The algorithms of score-based models can be connected to DDPMs, but are derived in a different way. Therefore,  it is possible that score-based models could also benefit from the learned prior idea, while not so straightforward in terms of applying our current ELBO-based framework to derive the corresponding algorithms. Nevertheless, we hope this work on improving conditional DDPMs with learnable priors can pave the way for answering the questions regarding applying the idea to continuous-time diffusion models in the future.

---

> ### Comment · Reviewer_Z1ga · 2024-11-25
>
> Thanks for your response, especially for clarifications regarding $\sigma_t$ and $\gamma_t$, now I see that everything is consistent in the theoretical part of the submission.

---

> ### Comment · Reviewer_Z1ga · 2024-11-26
>
> Despite some of my concerns have been addressed properly in the revised version of the paper, I've decided to keep my initial score unchanged. First of all, I think that the experiments related to W1 are quite important and should at least be mentioned and discussed in the main part of the paper rather than just in appendices, and maybe more detailed comparison for different tasks could be made. Also, I've carefully read discussions related to other reviews and I tend to partly agree with the comments of Reviewers WuzF and VRxE on the novelty.

---

> ### Author Response · Authors · 2024-11-27
> **Official Comment by Authors**
>
> We sincerely thank you for your time and effort, and for keeping the positive score on our work. We are glad to see that our clarifications to your initial questions are helpful, which have made the theoretical part more consistent in our paper. We are also grateful for your further suggestion to highlight the experiment related to **W1** in the main text rather than put in Appendix. **Accordingly, we have uploaded a revised PDF to reflect this change at lines 287 -- 295 and line 404--413 in the main text.** We thank the reviewer for the comment on more detailed comparison and will continue to expand our experimental results on different tasks going forward. In addition, we are happy to answer more questions and concerns that might come up before the end of the discussion period, and to provide further clarification to more specific aspects that the reviewer thinks could be helpful for the discussion on the novelty part.

---

> ### Author Response · Authors · 2024-12-02
> **Official Comment by Authors before extended discussion period closes**
>
> We once again thank the reviewer for their constructive comments and feedback that have helped us greatly improve the paper. We kindly remind the reviewer that there are only two days left before the closure of the discussion period, and we are happy to answer any further questions before that. We also sincerely invite the reviewer to look at our latest general response at ["Official Comment by Authors before End of Discussion Period"](https://openreview.net/forum?id=UbMYhX60tY&noteId=i2YnkJlbL7) if you have not had a chance, where we have provided more detailed and explicit comparison with existing approaches. We hope you find the discussion there helpful and interesting, and that it makes you more convinced about our innovation. Otherwise, we are happy to further address any specific concern during the remaining time for discussion. **We also managed to obtain more results related to your comment **W1** and would like to share them with you (please see below).**
>
> ---
> **Additional experimental results:**
>
> Besides the SE experiments related to your comment **W1**, we were able to also conduct the experiments with regards to using or not using the Posterior Net on the IR task on the RainDrop dataset. Due to insufficient time for completing the full experiments, we provided preliminary results of the models trained after 2,778 epochs as a reference in the table below. We compared RestoreGrad with and without Posterior Net for a few values of regularization weight $\eta$. The results again indicate the advantages of using the posterior encoder. We assure that the fully-trained model comparison will be included in the revised paper.
>
> **Table:** *IR comparison of RestoreGrad models trained with and without Posterior Net. The better results with Posterior Net again indicate that the posterior encoder is helpful.*
>
> | SE model                                       | PSNR $\uparrow$   | SSIM $\uparrow$    | LPIPS $\downarrow$ | FID $\downarrow$  |
> |------------------------------------------------|-------------------|--------------------|--------------------|-------------------|
> | RainDropDiff                                   | 29.94             | 0.9332             | 0.1489             | 27.62             |
> | + RestoreGrad                                  | **32.41**    | *0.9421* | **0.1409**    | **22.16**    |
> | + RestoreGrad w/o Posterior Net ($\eta=0$)     | 31.93             | 0.9356             | 0.1514             | 25.75             |
> | + RestoreGrad w/o Posterior Net ($\eta=0.005$) | 32.22             | 0.9416             | *0.1429* | *22.77* |
> | + RestoreGrad w/o Posterior Net ($\eta=0.01$)  | *32.29* | **0.9426**    | 0.1473             | 23.42             |
>
> *Best and second best values are indicated with bold text and  italicized text, respectively.

---

### Official Review · Reviewer_WuzF · 2024-11-04

**Soundness:** 2
**Presentation:** 3
**Contribution:** 2
**Rating:** 5
**Confidence:** 5

**Summary:**

This paper proposes RestoreGrad, a framework that enhances signal restoration tasks using conditional denoising diffusion probabilistic models (DDPMs) with a jointly learned prior. The authors introduce a data-dependent prior distribution to improve the efficiency of DDPMs for recovering clean signals from degraded versions. RestoreGrad’s framework incorporates two additional networks, Prior Net and Posterior Net, to capture more relevant information for the reverse diffusion process.

**Strengths:**

1. Adaptive Prior Distribution: RestoreGrad learns a data-dependent prior rather than relying on a fixed Gaussian, potentially improving efficiency by tailoring the prior to the degraded signal.

2. Joint Training with Minimal Modifications: The proposed approach requires only the addition of Prior and Posterior networks and a modified loss function, making it compatible with various DDPM architectures without significant changes.

3. Efficiency in Training and Inference: RestoreGrad exhibits faster convergence and robustness to fewer sampling steps during inference, making it computationally efficient compared to baseline DDPMs.

**Weaknesses:**

1.  Lack of Novelty in Methodology: The method heavily builds on existing techniques, including the use of conditional DDPMs, adaptive priors (as seen in PriorGrad), and standard VAE and DDPM loss formulations. The Prior and Posterior networks adapt concepts from encoder-decoder architectures, similar to those in VAEs, rather than presenting new model designs.

2. Learned Prior Concept is not novel: The idea of using an adaptive, learned prior is inspired by existing models, like PriorGrad [1]. PriorGrad introduced the concept of data-dependent priors by using the degraded signal to inform the prior distribution, so RestoreGrad's approach of a learned prior based on the degraded input builds directly on this idea rather than introducing a fundamentally new approach
3. Limited Novelty in Loss Function Design: The modified ELBO loss primarily integrates known VAE and DDPM loss terms without introducing novel loss structures or constraints tailored to signal restoration.

4. No New Restoration-Specific Techniques: The Prior and Posterior networks are based on standard architectures (e.g., ResNet), and there is no exploration of restoration-specific methods, such as task-specific losses or architectures, that might further enhance the model's performance in different restoration tasks.

5. Missing Computational Time Comparison: The paper lacks a clear and detailed computational time comparison with baseline models, which is critical for understanding the added cost of the Prior and Posterior Nets. Including this comparison in training and inference steps, along with performance metrics, would provide a clearer assessment of RestoreGrad's computational demands.

6. Missing FID details in Tables 3, 4, and 7 - While the paper reports PSNR and SSIM metrics for assessing image quality, it omits the FID (Fréchet Inception Distance) metric, a widely used measure for evaluating the realism and distributional similarity of generated images. Including FID would give a more comprehensive understanding of RestoreGrad's performance in terms of perceptual quality.

7. No Discussion on Trade-offs: The method introduces additional computational layers and complexity, yet there's little discussion on the trade-offs between quality improvement and increased model size or computation.

[1]. Lee, S. G., Kim, H., Shin, C., Tan, X., Liu, C., Meng, Q., ... & Liu, T. Y. (2021). Priorgrad: Improving conditional denoising diffusion models with data-dependent adaptive prior. arXiv preprint arXiv:2106.06406.

**Questions:**

How the authors address the above weakness will be important for re-evaluating the paper.

---

> ### Author Response · Authors · 2024-11-22
> **Response to Reviewer WuzF (W1~W2)**
>
> ## Weaknesses
>
> > **W1:** Lack of Novelty in Methodology: The method heavily builds on existing techniques, including the use of conditional DDPMs, adaptive priors (as seen in PriorGrad), and standard VAE and DDPM loss formulations. The Prior and Posterior networks adapt concepts from encoder-decoder architectures, similar to those in VAEs, rather than presenting new model designs.
>
> We are grateful for the reviewer's feedback and would like to provide further clarification. It is true that our method is inspired by several existing techniques in generative modeling, spanning form the classical VAE to the most recent and powerful DDPM. However, the way we connect these techniques to improve upon current diffusion-based methods in a **parameter-efficient**, **easily generalizable** manner to the **signal restoration** problems, is original and has not been established before.
>
> In this paper, we formally derive our loss function and learning framework by integrating the ELBOs of VAEs and DDPMs as presented in eq. (7) of the original manuscript. **To our best knowledge, this derivation is original and has not been done before.** Furthermore, based on the observation of the correlation between the noisy and clean signals, we propose to adopt the two-encoder training scheme as in Figure 4. Note that, using two encoders is not new in the literature. However, **utilizing prior and posterior encoders in the context of DDPM-based signal restoration to encode the correlation between noisy and clean signals is novel.**
>
> In terms of the difference to the VAE and DDPM, the proposed framework provides a new angle of improvements to the two methodologies, enjoying the merits of the two types. Typically, the VAE employs an encoder-decoder architecture while DDPM is a decoder-only, iterative method. Our framework seamlessly combines the two in a formal manner. From the VAE perspective, we replace conventional decoders in VAEs by an iterative decoder with **greater generation capabilities**. From the DDPM point of view, we equip the diffusion model with encoders to extract more informative representations for **improved model efficiency**.
>
> In contrast to PriorGrad that has studied adaptive (i.e., data-dependent) priors, we further explore feasible improvements by systematically **learning** the adaptive prior. Our framework also aims at the applications where **PriorGrad might not be optimal**, i.e., signal restoration, as PriorGrad estimates the prior only based on using the degraded signal.
>
> Finally, as mentioned above, the utilization of prior and posterior encoders with the DDPM model for signal restoration is one of our contributions. Since there is no existing work on such DDPM framework design, we adopted standard architectures (ResNets) for the encoder modules **for simplicity**.  This adoption of in turn demonstrates that the proposed learning framework is **flexible** and **generalizable** since the idea simply works by using standard architectures.
>
> > **W2:** Learned Prior Concept is not novel: The idea of using an adaptive, learned prior is inspired by existing models, like PriorGrad [1]. PriorGrad introduced the concept of data-dependent priors by using the degraded signal to inform the prior distribution, so RestoreGrad's approach of a learned prior based on the degraded input builds directly on this idea rather than introducing a fundamentally new approach.
>
> Thank you for your comment and we would like to make some more clarification. First, PriorGrad only relies on the condition signal $\mathbf{y}$ to compute the data-dependent prior. **Therefore, PriorGrad is not optimal for the applications where the condition signal contains certain degradations, e.g., in speech enhancement where $\mathbf{y}$ is a noisy version of the target $\mathbf{x}_0$**. This poses potential issues, since the computed prior might not be closer to the target signal distribution but to the noisy signal. In contrast, our RestoreGrad exploits the target signal $\mathbf{x}_0$ information in addition to $\mathbf{y}$ for prior estimation.
> Moreover, **PriorGrad actually failed in demonstrating the benefits of using learnable priors, and only settled on handcrafted priors**. In fact, **a formal learned prior framework is not explicitly discussed in PriorGrad**.
> On the contrary, in RestoreGrad we formally formulate the jointly learnable prior idea and make the idea actually work and applicable to signal restoration problems, which we believe deserve being considered as novel contributions as we address the underlying issues not sufficiently discussed in existing works on data-dependent priors.

---

> ### Author Response · Authors · 2024-11-22
> **Response to Reviewer WuzF (W3~W5)**
>
> > **W3:** Limited Novelty in Loss Function Design: The modified ELBO loss primarily integrates known VAE and DDPM loss terms without introducing novel loss structures or constraints tailored to signal restoration.
>
> Thank you for the comment. The formal integration of ELBOs of VAEs and DDPMs as shown in eq. (7) of the old manuscript, to our knowledge, is original. More specifically, VAE utilizes a latent variable $\boldsymbol{\epsilon}$ in its ELBO, arriving at an encoder-decoder architecture, while DDPM employs a sequence of latent variables $\mathbf{x}_{1:T}$ and results in a decoder-only, iterative method. Our framework, by introducing the second inequality in eq. (7), merges the two approaches into a better one, and embraces the best of the two worlds -- generation capabilities and model efficiency. Furthermore, although the resulting loss function seems a direct combination of VAE and DDPM, **it is formally derived from a modification to the ELBOs**, rather than just a heuristic integration of known loss terms from both sides. Moreover, we propose the usage of prior and posterior encoders for realizing the learning objective. This design option is based on the observation that the noisy signal usually correlates with the clean signal to a certain level in signal restoration problems, making it more useful and specific for such applications.
>
> > **W4:** No New Restoration-Specific Techniques: The Prior and Posterior networks are based on standard architectures (e.g., ResNet), and there is no exploration of restoration-specific methods, such as task-specific losses or architectures, that might further enhance the model's performance in different restoration tasks.
>
> We thank the reviewer for their opinion on task-specific techniques. We understand that the reviewer might be seeking a bit more sophisticated modifications to the model architectures or loss functions customized to different signal restoration tasks, which makes sense to us. However, we would also like to point out if our framework had to rely on task-specific techniques, its generalizability could be downgraded at the same time. Indeed, generalizability to different modalities and tasks is one of our advantages. Thus, **demonstrating that our framework simply works with standard architectures like the ResNet suggests its effectiveness and generality, and should be considered as a strength, not a weakness**. As the reviewer pointed out, the model's performance could be further enhanced by using more task-specific losses or architectures. We agree with that, and are hoping our prior learning framework utilizing standard architectures can serve as a good baseline for future research works on more task-specific improvements for speech, image, or other modalities.
>
> > **W5:** Missing Computational Time Comparison: The paper lacks a clear and detailed computational time comparison with baseline models, which is critical for understanding the added cost of the Prior and Posterior Nets. Including this comparison in training and inference steps, along with performance metrics, would provide a clearer assessment of RestoreGrad's computational demands.
>
> Thank you for the suggestion. We have added the required training time and the corresponding device type to the experimental setup sections (in Section 5.1.1 (SE) and Section 5.2.1 (IR) of the revised manuscript) of the models for computation time information. In our experiments we observed, for the same number of total epochs, the baseline DDPM model and the RestoreGrad model took similar training time, as the additional parameters introduced by Prior and Posterior Nets are significantly fewer than the DDPM model itself. Thus, given that RestoreGrad could potentially achieve better performance over the DDPM with much fewer training steps, it requires fewer GPU hours or days to learn a model with stronger performance.
> For example, from Figure 2 we observe roughly $4\times$ reduction of training epochs for the RestoreGrad to achieve better performance than the baseline diffusion model. This indicates a saving of 9 GPU days when training the model on two NVIDIA Tesla V100 GPUs of 32 GB CUDA memory for the IR task on the RainDrop dataset (Note that the models were trained on less powerful GPUs (2 NVIDIA Tesla V100s) so the total training time seems quite long. With more advanced GPUs it could be considerably reduced). The key takeaway message is that, with equivalent hardware conditions, RestoreGrad can learn a diffusion model with better performance by using much fewer training iterations than the baseline DDPM.

---

> ### Author Response · Authors · 2024-11-22
> **Response to Reviewer WuzF (W6~W7)**
>
> > **W6:** Missing FID details in Tables 3, 4, and 7 - While the paper reports PSNR and SSIM metrics for assessing image quality, it omits the FID (Fréchet Inception Distance) metric, a widely used measure for evaluating the realism and distributional similarity of generated images. Including FID would give a more comprehensive understanding of RestoreGrad's performance in terms of perceptual quality.
>
> We thank the reviewer for suggesting that we include FID results to facilitate understanding of our approach in terms of perceptual image quality. To account for the missing details, we have i) replaced the PSNR and SSIM plots in Figure 2 of the revised PDF by the FID and LPIPS plots (i.e., Figure 13 in the original manuscript) and ii) added FID and LPIPS numbers to Table 7 and Table 13 of the revised PDF to make the comparison more comprehensive. Note that we did not include them to Table 3 and Table 4 as the SOTA methods did not provide evaluation results of FID and LPIPS.
>
> > **W7:** No Discussion on Trade-offs: The method introduces additional computational layers and complexity, yet there's little discussion on the trade-offs between quality improvement and increased model size or computation.
>
> Thank you for mentioning the discussion on the trade-offs between quality and complexity. In this work, we adopted Prior Net and Posterior Net that are much smaller compared to the corresponding DDPM model in each task. The increased computation is thus almost negligible as RestoreGrad took similar training time as training the baseline DDPM. Per the reviewer's suggestion, we have further conducted experiments on using different model sizes for the Prior and Posterior Nets for the SE task. The results in the table below clearly show that the restored speech quality improves with an increased model size of the encoder(s). We have included the relevant discussion to the revised manuscript (Appendix C.1, Table 11).
>
>
> **Table:** *SE comparison of RestoreGrad models using three different sizes of the encoder modules (i.e., Prior Net and Posterior Net). The Base (96K) model is the one used in main experiments.*
>
> | Encoder size        | PESQ $\uparrow$ | COVL $\uparrow$ | SSNR $\uparrow$ | SI-SNR $\uparrow$ |
> |---------------------|-----------------|-----------------|-----------------|-------------------|
> | Tiny (24K params)   | 2.48            | 3.11            | 5.10            | 13.74             |
> | Base (96K params)   | 2.51            | 3.14            | 5.92            | 14.74             |
> | Large (370K params) | 2.54            | 3.16            | 6.15            | 15.01             |

---

> ### Author Response · Authors · 2024-12-02
> **Official Comment by Authors before extended discussion period closes**
>
> We once again thank the reviewer for their time and effort. Given the approaching discussion deadline, we hope they had a chance to look at our response and let us know if we have addressed their concerns. For the sake of convenience we provide a summary below:
>
> - We provided clarification on the novel aspects of our work, motivation of our chosen architectures, and differentiation from existing methods, including a more explicit, side-by-side comparison at our latest general response: ["Official Comment by Authors before End of Discussion Period"](https://openreview.net/forum?id=UbMYhX60tY&noteId=i2YnkJlbL7).
> - We provided new experimental results on the model size and quality improvement trade-offs, for both the SE ([see results in Table here](https://openreview.net/forum?id=UbMYhX60tY&noteId=afzQMeVI8n)) and IR ([see results in Table here](https://openreview.net/forum?id=UbMYhX60tY&noteId=uqEXjCW6EJ)) tasks.
> - We included evaluation with more quality metrics and information of computational time in the experimental sections of our revised manuscript.
>
> Given that we have addressed all the questions raised by the reviewer by extensive analysis and new experiments, we hope the reviewer is satisfied with the answers. Otherwise, please feel free to ask any questions during the next two days of the discussion period. Thank you again for your time and effort, we greatly appreciate it.

---

### Author Response · Authors · 2024-11-22
**General Response to All Reviewers**

We thank all the reviewers for the insightful comments that have helped us improve our submission substantially. We were able to conduct additional experiments to address reviewers’ feedback and concerns, further improve the manuscript clarity, and justify the contributions and innovation of RestoreGrad. Here we provide an overview of our responses. Detailed responses can be found in the reviewers' individual comments. We have also updated the manuscript accordingly, highlighting the changes in blue for ease of review.

---

## Paper summary

- We study improvements to conditional denoising diffusion probabilistic models (DDPMs) for signal restoration tasks, where clean signals are recovered from their degraded observations by utilizing the generative modeling capabilities of diffusion models.

- To address inefficiencies due to the discrepancy between the actual data and assumed prior distributions used in existing DDPMs, we propose a new framework called RestoreGrad that jointly learns the prior distribution with the diffusion model through a *prior encoder* for improved model efficiency, based on exploiting the correlation between the targe clean signal and input degraded signal encoded by an auxiliary *posterior encoder*. The framework is formally derived from a novel integration of the VAE and DDPM.

- We show that the new, two-encoder DDPM learning framework leveraging jointly learned priors leads to faster convergence, better restored signal quality, and robustness to using fewer sampling steps during inference over existing DDPM baselines on speech and image restoration tasks.

---

## Strengths as per the reviewers

- Reviewers WuZF, RjT2 appreciated the generalizability and effectiveness of the proposed work. Specifically, they highlighted the significance of the proposed idea being applicable to many techniques of similar frameworks, without requiring complicated modifications to the diffusion models and loss functions.

- Reviewers VRxE, RjT2 acknowledged the parameter efficiency of our framework, as RestoreGrad achieves notable improvements while introducing only minimum additions of parameters over baseline conditional DDPMs in signal restoration tasks.

- Reviewer Z1ga appreciated the idea of combining techniques from VAEs and DDPMs via a novel ELBO-based loss function to arrive at the two-encoder design for jointly learning the prior. Reviewer VRxE acknowledged the robustness to sampling using fewer timesteps as an important feature of RestoreGrad. Reviewer RjT2 highlighted the performance of RestoreGrad not being sensitive to the values of introduced hyper-parameters as a significant advantage for real-world applications.

---

## Major questions from the reviewers

While we respond to the individual queries in reviewer specific rebuttals, here we list some of the important ones and our responses.

> **Distinction with existing approaches**

Reviewers WuzF and VRxE had some questions on the novelty of the learned prior concept and distinction with existing works on adaptive and learned priors (e.g., PriorGrad and Grad-TTS). We summarize our main novel points here.

- Our framework adopts a **two-encoder** design, i.e., prior and posterior encoders, for **jointly** learning the diffusion prior distribution by incorporating posterior information based on the correlation between the target clean signal and degraded observation(s). Such a network design of using prior and posterior encoders for DDPMs has not been seen in the literature to our knowledge.

- Our learning framework is established based on **a novel loss function** formally derived from integrating the ELBOs of VAEs and DDPMs to enjoy the advantages of both methodologies. The derivation and resulting loss function are both new.

- Our proposed method is general and can be adopted by DDPM-based signal restoration models for several data modalities including images and audio. The **generality** of our approach further distinguishes it from prior works that focused on just speech.

> **Completeness of experiments**

The reviewers have several questions on missing detailed information of the experiments and also suggestions on additional results. We have revised our manuscript accordingly to provide more complete evidence. Below we summarize main additional experiments conducted **in the revised manuscript**.
- Reviewer Z1ga suggested an interesting ablation study to observe the benefits brought by including the posterior information coming from the target signal $\mathbf{x}\_0$ and conditioning signal $\mathbf{y}$ through the Posterior Net. We presented the new results in Section 5.1.2, **Table 3**.

- Reviewer VRxE suggested a more perceptual SE metric using ASR. We added the results using open-source ASR models in Appendix C.1, **Table 10**.

- Reviewer WuzF asked about the trade-offs between quality and complexity. We added related discussion by showing the performance versus encoder model size in Appendix C.1, **Table 11**.

---

### Author Response · Authors · 2024-11-29
**Official Comment by Authors before End of Discussion Period (1/2)**

We thank the AC for their efforts so far in facilitating reviews and their efforts in the future during the rest of the conference. We would also like to thank all the reviewers again for their time and effort on providing valuable feedback during the discussion period by far. We are glad that by addressing the reviewers' concerns and questions, our paper has apparently improved. It was especially encouraging to see that **Reviewers VRxE** and **RjT2** have increased their scores to reflect more positive recognition of our work, and **Reviewer Z1ga** stays with their initial positive view. On the other hand, *it was unfortunate that we have not heard back from **Reviewer WuzF** so far* after we responded to their initial review comments, who had concerns about the novelty of RestoreGrad. We therefore would like to take this chance to further highlight our novel aspects and distinctions from the most similar prior works, from both the Algorithm and Application perspectives. We hope this helps you better appreciate the originality and significance of our work.

## Originality on the Algorithm Side

### **1. New Evidence Lower Bound (ELBO) on Conditional Data Log-likelihood for Generative Modeling**

Let us compare the ELBOs used in different generative modeling frameworks:

- Conditional VAEs (e.g., Refs [R1, R2]):
$$
\\log p(\\mathbf{x}\_0|\\mathbf{y})\\geq\\mathbb{E}\_{q\_\\phi(\\boldsymbol{\\epsilon}|\\mathbf{x}\_0,\\mathbf{y})}\\left[\\log p\_\\theta(\\mathbf{x}\_0|\\mathbf{y},\\boldsymbol{\\epsilon})\\right] - D\_{\\text{KL}}\\left(q\_\\phi(\\boldsymbol{\\epsilon}|\\mathbf{x}\_0,\\mathbf{y}) || p\_\\theta(\\boldsymbol{\\epsilon}|\\mathbf{y})\\right).
$$

- Conditional DDPMs (e.g., Refs [R3, R4, R5]):
$$
\\log p(\\mathbf{x}\_0|\\mathbf{y})\\geq\\mathbb{E}\_{q(\\mathbf{x}\_{1:T}|\\mathbf{x}\_0)}\\left[\\log\\frac{p\_\\theta({\\mathbf{x}\_{0:T}}|\\mathbf{y})}{q(\\mathbf{x}\_{1:T}|\\mathbf{x}\_0)}\\right].
$$

- RestoreGrad (ours):
$$
\\log p(\\mathbf{x}\_0|\\mathbf{y})\\boldsymbol{\\geq}\\mathbb{E}\_{q_\\phi(\\boldsymbol{\\epsilon}|\\mathbf{x}\_0,\\mathbf{y})}\\Biggr[\\mathbb{E}\_{q(\\mathbf{x}\_{1:T}|\\mathbf{x}\_0)}\\left[\\log\\frac{p\_{\\theta}(\\mathbf{x}\_{0:T}|\\mathbf{y},\\boldsymbol{\\epsilon})}{q(\\mathbf{x}\_{1:T}|\\mathbf{x}\_0)}\\right]\\Biggr] - D\_{\\text{KL}}\\bigr(q\_{\\phi}(\\boldsymbol{\\epsilon}|\\mathbf{x}\_0,\\mathbf{y}) || p\_{\\psi}(\\boldsymbol{\\epsilon}|\\mathbf{y})\\bigr).
$$

We can see that our method presents *a new form of ELBO that seamlessly integrates the DDPM into the VAE framework.* The motivation behind the combination is to embrace the best of both worlds, i.e., promising generative ability (DDPM) and modeling efficiency (VAE). Consequently, RestoreGrad achieves improved output signal quality and training/sampling efficiency *simultaneously*.

### **2. New Loss Function for Conditional DDPM Training**

The closest work to ours is PriorGrad (Ref [R6]) who also improved conditional DDPMs with data-dependent priors. Motivated by their adaptive prior concept, our framework further generalizes to *a jointly learnable prior framework by employing a prior encoder and a posterior encoder, $\psi$ and $\phi$.* To be more specific, we can compare the loss functions of ours with PriorGrad:
- PriorGrad (Ref [R6]):
$$
\\min\_{\\theta} \\,\\,\\, ||\\boldsymbol{\\epsilon}-\\boldsymbol{\\epsilon}\_\\theta(\\mathbf{x}\_t,\\mathbf{y},t)||^2\_{\\boldsymbol{\\Sigma}^{-1}\_{y}}, \\,\\,\\, \\text{where} \\,\\,\\, \\boldsymbol{\\Sigma}\_{y}=f(\\mathbf{y}) \\, \\text{and} \\, f(\cdot) \\, \\text{is a pre-defined function that maps} \\, \\mathbf{y} \\, \\text{into the prior distribution.}
$$

-  RestoreGrad (ours):
$$
\\min_{\\theta,\\phi,\\psi} \\,\\,\\, \\eta\\bigr(\\bar{\\alpha}\_T||\\mathbf{x}\_0||^2\_{\\boldsymbol{\\Sigma}^{-1}\_{\\text{post}}}+\\log|\\boldsymbol{\\Sigma}\_{\\text{post}}|\\bigr)+||\\boldsymbol{\\epsilon}-\\boldsymbol{\\epsilon}\_\\theta(\\mathbf{x}\_t,\\mathbf{y},t)||^2\_{\\boldsymbol{\\Sigma}^{-1}\_{\\text{post}}}+\\lambda \\bigr(\\log\\frac{|\\boldsymbol{\\Sigma}\_{\\text{prior}}|}{|\\boldsymbol{\\Sigma}\_{\\text{post}}|}+\\text{tr}(\\boldsymbol{\\Sigma}\_{\\text{prior}}^{-1}\\boldsymbol{\\Sigma}\_{\\text{post}})\\bigr), \\,\\,\\, \\text{where} \\,\\,\\, \\boldsymbol{\\Sigma}\_{\\text{prior}}\\coloneqq\\boldsymbol{\\Sigma}\_{\\text{prior}}(\\mathbf{y};\\psi) \\,\\,\\, \\text{and} \\,\\,\\,
\\boldsymbol{\\Sigma}\_{\\text{post}}\\coloneqq\\boldsymbol{\\Sigma}\_{\\text{post}}(\\mathbf{x}\_0,\\mathbf{y};\\phi).
$$

We can see that our framework is more *generalized* as it takes care of the prior estimation by utilizing two encoders $\psi$ and $\phi$ to jointly learn the distribution with the DDPM $\theta$, therefore by-passing the requirement to search for a suitable mapping function $f(\cdot)$ usually having to be based on the domain knowledge of a given task (e.g., spectral analysis in speech tasks). Our framework is thus more general.

---

> ### Author Response · Authors · 2024-11-29
> **Official Comment by Authors before End of Discussion Period (2/2)**
>
> ## Novelty on the Application Side
>
> ### **1. Efficient DDPM Training and Sampling for High Quality Signal Restoration**
>
> We are the first to apply the jointly learnable prior idea in DDPMs to *Signal Restoration* tasks and demonstrate improved efficiency of using jointly learned priors in both training and sampling.
>
> ### **2. Generalizability to Different Modalities**
>
> RestoreGrad is quite flexible for applying to various signal types, as it is based on the general observation that the target clean signal and the input noisy signal are correlated to certain degrees. In this paper, we demonstrate RestoreGrad in the speech and image domains, while previous works of data-dependent priors (e.g., PriorGrad (Ref [R6]), Grad-TTS (Ref [R7])) only focused on speech tasks and thus are much more specific than ours.
>
> ### **3. Considerable Improvements with Lightweight Modules**
>
> We show that our framework achieves significant improvements by using standard, lightweight architectures (ResNets) for the introduced encoder modules (Prior and Posterior Nets). Being able to economically attain such improvements further supports the advantages and effectiveness of RestoreGrad.
>
> ### We sincerely wish that you find the above discussion helpful and interesting; and, reinforce your positive view of the paper and make you more supportive of our work. We are also happy to answer any further questions and comments from the reviewers anytime before the discussion period ends.
>
> **Refs:**
>
> - [R1] Esser et al., "A variational U-Net for conditional appearance and shape generation," CVPR, 2018.
>
> - [R2] Harvey et al., "Conditional image generation by conditioning variational auto-encoders," ICLR, 2022.
>
> - [R3] Ozdenizci and Legenstein, "Restoring vision in adverse weather conditions with patch-based denoising diffusion models," IEEE TPAMI, 2023.
>
> - [R4] Croitoru et al,, "Diffusion models in vision: A survey,"  IEEE TPAMI, 2023.
>
> - [R5] Cao et al., "A survey on generative diffusion models," IEEE TKDE, 2024.
>
> - [R6] Lee et al., "PriorGrad: Improving conditional denoising diffusion models with data-dependent adaptive prior," ICLR, 2021.
>
> - [R7]  Popov et al., "Grad-TTS: A diffusion probabilistic model for text-to-speech," ICML, 2021.
>
> ---
>
> ### **Additional results on complexity and quality improvement trade-offs**
>
> We were able to obtain image restoration (IR) results related to the comment (**W7**) asked by **Reviewer WuzF** on the trade-offs between complexity and quality improvement in addition to the speech enhancement (SE) results we provided earlier in the individual response to the reviewer. To be more specific, we trained and evaluated another model of RestoreGrad on the RainDrop IR dataset (Ref [R8]) with a larger encoder model (1.9M params) than the originally used 0.27M model. The results are presented in the table below, where we can again observe that a larger encoder leads to improved restoration quality. This is in consistence with our previous observation from the SE results.
>
> **Table:** *IR comparison of RestoreGrad models using a larger size of the encoder modules (i.e., Prior Net and Posterior Net) on RainDrop dataset. We show PSNR/SSIM scores vs. training epochs for both models.* We see that a larger encoder leads to improved performance consistently across different training epochs.
>
> | Encoder size / Epochs | @2.3K                          | @4.6K                          | @5.6K                          | @8.3K                          | @9.3K                          |
> |-----------------------|--------------------------------|--------------------------------|--------------------------------|--------------------------------|--------------------------------|
> | Base (0.27M params)   | 32.25/0.9414                   | 32.49/0.9432                   | 32.53/0.9435                   | 32.61/0.9439                   | 32.65/0.9441                   |
> | Large (1.9M params)   | **32.34**/**0.9418** | **32.60**/**0.9433** | **32.65**/**0.9436** | **32.69**/**0.9441** | **32.77**/**0.9444** |
>
> *Better values are indicated with bold text.
>
> **Refs:**
>
> - [R8] Qian et al., "Attentive generative adversarial network for raindrop removal from a single image," CVPR, 2018.

---

### Meta-Review · Area_Chair_UQe8 · 2024-12-21

**Metareview:**

This paper proposed a framework called RestoreGrad to deal with the signal restoration tasks using conditional diffusion models with a jointly learned prior. The key idea of RestoreGrad is to integrate VAE and DDPM to better capture the correlation between clean and degraded signals.

Strengths:

1. The integration of VAE and DDPM into a single  framework is interesting.

2. Compared to baseline models,  RestoreGrad demonstrates faster convergence and improvement with fewer sampling steps.


Weaknesses:
1. The proposed framework is similar to PriorGrad which makes the technical contribution marginal.

2. The comparisons is insufficient, which include some old baselines for image restoration tasks.

3.  More discussion is needed on computational complexity versus performance trade-offs.

4.  The writing is not very clear, e.g., the role of the posterior encoder and the derivation of the ELBO.

While this is an interesting paper, based on the above weaknesses raised by reviewers,  I have to recommend rejection in its current form but I sincerely encourage  the authors to revise it for future submission.

**Additional Comments On Reviewer Discussion:**

In the rebuttal, the authors provided clarifications but did not fully address address the concerns, e.g., marginal novelty, outdated baselines, analysis of computational cost versus quality improvements.

---

### Decision · Program_Chairs · 2025-01-22

Reject